# Implications of the 2019–2020 megafires for the biogeography and conservation of Australian vegetation

Robert C. Godfree [1✉], Nunzio Knerr[1], Francisco Encinas-Viso [1], David Albrecht[2], David Bush [3], D. Christine Cargill [2], Mark Clements[2], Cécile Gueidan [1], Lydia K. Guja[2], Tom Harwood[4], Leo Joseph[5], Brendan Lepschi[2], Katharina Nargar [6], Alexander Schmidt-Lebuhn [1] & Linda M. Broadhurst [1]

Australia's 2019–2020 'Black Summer' bushfires burnt more than 8 million hectares of vegetation across the south-east of the continent, an event unprecedented in the last 200 years. Here we report the impacts of these fires on vascular plant species and communities. Using a map of the fires generated from remotely sensed hotspot data we show that, across 11 Australian bioregions, 17 major native vegetation groups were severely burnt, and up to 67–83% of globally significant rainforests and eucalypt forests and woodlands. Based on geocoded species occurrence data we estimate that >50% of known populations or ranges of 816 native vascular plant species were burnt during the fires, including more than 100 species with geographic ranges more than 500 km across. Habitat and fire response data show that most affected species are resilient to fire. However, the massive biogeographic, demographic and taxonomic breadth of impacts of the 2019–2020 fires may leave some ecosystems, particularly relictual Gondwanan rainforests, susceptible to regeneration failure and landscape-scale decline.

[1] Centre for Australian National Biodiversity Research, CSIRO National Research Collections Australia, Canberra, ACT, Australia. [2] Centre for Australian National Biodiversity Research, Australian National Botanic Gardens, Canberra, ACT, Australia. [3] Australian Tree Seed Centre, CSIRO National Research Collections Australia, Canberra, ACT, Australia. [4] CSIRO Land and Water, Canberra, ACT, Australia. [5] Australian National Wildlife Collection, CSIRO National Research Collections Australia, Canberra, ACT, Australia. [6] Australian Tropical Herbarium, James Cook University, Cairns, QLD, Australia. ✉email: Robert.Godfree@csiro.au

There is emerging evidence from ecosystems worldwide that catastrophic events such as extreme drought and large bushfires can push terrestrial ecosystems past tipping points that result in abrupt ecosystem change[1,2]. Given the impact of human-driven climate change on the frequency and intensity of these events there is a need to quantify their effects on plant and animal communities[3] as they unfold. During the spring and summer of 2019–2020 south-eastern Australia experienced a severe bushfire season (the 'Black Summer') during which millions of hectares of natural vegetation along the eastern coast[4], much of which had already been exposed to prolonged drought and record high temperatures (Supplementary Fig. 1a, b), were burnt. The area burned was almost an order of magnitude larger than other major global fires of the past decade[5]. Given the magnitude of this event we now need to identify and prioritise conservation and recovery actions across the fire-affected areas.

In this paper, we quantify the biogeographic and taxonomic impact of the 2019–2020 bushfires on vascular plant taxa and associated vegetation types across the south-east of the Australian mainland using a continent-wide fire layer and geocoded species occurrence data. Our findings show that the megafires extensively burnt a broad range of vegetation communities, including 72–83% of rainforests, eucalypt forests and woodlands, and shrublands and heathlands within individual subcontinental-scale bioregions. Importantly, the megafires occurred within globally significant biodiversity hotspots with high richness and endemism across important plant groups (e.g., Proteaceae[6], eucalypts), including Gondwanan and subtropical rainforest species. Our assessment estimates >800 vascular plants had >50% of their populations or ranges burn. Although trait analysis reveals that most of the plants affected are likely fire resilient, the results show potentially important effects on the demography and viability of many plant species across multiple taxonomic groups.

## Results and discussion

**Extent of fires in vegetation communities**. We developed a continent-wide fire layer based on Digital Earth Australia remotely sensed historical hotspot data gridded at a 2.5 km resolution over the period 1 July 2019 to 11 February 2020 (Fig. 1a; see 'Methods'). Based on this layer we estimate that some 200 major fires burnt through 10.4 million hectares (Mha; ca. 25.7 million acres) of land in south-eastern Australia (Fig. 1b) until extinguished or brought under control in mid-February 2020. The fires progressed from north to south and grew most rapidly between September 2019 and January 2020 (Supplementary Fig. 2). Very large fires occur regularly in south-eastern Australia, but the aggregate spatial extent of the Black Summer bushfires in the states of Victoria and New South Wales (NSW) doubles major fire events of recent decades (all ca. 0.5–5 Mha; Supplementary Data 1), and even the 1851 Victorian fires of 'Black Thursday' (ca. 5 Mha). A significant feature of the event was the massive size of individual fires: eleven exceeded 0.1 Mha (i.e., 100,000 ha) and seven exceeded 0.40 Mha (ca. 1 million acres; Fig. 1b). Most megafires (here defined as >0.1 Mha) arose following the merging of multiple, independent large fires. Another six major fires (0.050–0.099 Mha) and many smaller fires also occurred across the region (Fig. 1b). The largest two megafires in northern NSW formed a complex approximately 380 km long that covered 2.66 Mha (6.57 million acres) (Fig. 1b), ranking as one of the largest contiguous fires ever recorded globally.

Approximately three-quarters of burnt areas occurred within remnant wooded vegetation (Fig. 2a, b). Among broad vegetation types based on Australian Major Vegetation Groups (MVG; see Table 1)[7], eucalypt forests and woodlands (7.34 Mha) were most affected, but large areas of rainforests and vine thickets (0.33 Mha;

Fig. 2C), shrublands and heathlands (0.22 Mha) and other forest and woodland ecosystems (0.25 Mha) were also burnt (Table 1). The biogeographic extent of the Black Summer fires is revealed by the fact that these broad vegetation types were impacted across the eleven south-eastern Australian bioregions (IBRA bioregions[8]) that occur within the study area (Fig. 1b and Supplementary Table 1). Among these, coastal and near-coastal bioregions (NSW North Coast, Sydney Basin, South East Corner and New England Tablelands) had the highest percentage of these vegetation types burn ($PF_T$), including 4.40 Mha ($PF_T = 59\%$) of eucalypt forests and woodlands, 0.25 Mha ($PF_T = 59\%$) of rainforest and vine thickets and 0.18 Mha ($PF_T = 69\%$) of shrublands and heathlands (Table 1). In some bioregions $PF_T$ exceed 75%, notably for rainforests in the New England Tablelands and the South East Corner of NSW (Fig. 1b and Supplementary Table 1). The remaining burnt areas mainly consisted of cleared or non-native vegetation (2.06 Mha).

Our data also indicate that hotspot temperature varied significantly within all major fires (Fig. 1b). Almost half of all grid cells (46%) contained a very high or high relative fire hotspot temperature ($T_R$; see 'Methods'). These often occurred along the western edges of the megafires, particularly in northern and central NSW (Fig. 1b). Conversely, unburnt areas >5000 ha in size were detected inside all megafires (Fig. 1b), usually surrounded by large areas of lower fire temperature. Analysis of a spatial impact metric, $I_H$, based on the estimated percentage of area in very high and high relative fire temperature categories ($T_R > 0.50$), predicts that 17 Australian Major Vegetation Groups[7] were severely affected, suffering from both extensive fires ($PF_T > 50\%$) and relatively high fire temperatures ($I_H > 25\%$, Table 1) in one or more bioregions. The most geographically extensive of these include Eucalypt Tall Open Forests (1.14 Mha across five bioregions), Eucalypt Open Forests (2.0 Mha across three bioregions), Eucalypt Woodland (0.39 Mha in one bioregion), Rainforests and Vine Thickets (0.22 Mha across three bioregions) and heathlands (0.16 Mha across four bioregions; Table 1). Again, coastal and near-coastal bioregions were most affected: here 10 MVGs each had >50% of their total area burn, and 15 experienced widespread hot fires ($I_H > 25\%$) in at least one bioregion (Table 1). Satellite images clearly demonstrate the dramatic landscape-level 'browning' that followed the Black Summer fires in these areas, with the loss of most of the wooded canopy leaving only a mosaic of smaller unburnt areas (Fig. 2d).

The scale of the fires and the breadth of vegetation types affected during the 2019–2020 fire season has implications for biodiversity conservation both in Australia and globally. Many of these ecosystems comprise a globally significant biodiversity hotspot[9,10] with high richness, rarity, endemism and phylodiversity evident among the Proteaceae[6], Asteraceae[11], conifers[12], eucalypts[13], bryophytes[14], and other plant groups[15,16]. The region also contains cool-temperate Gondwanan relictual and subtropical rainforest species that have suffered extensive clearing in past decades[9] and today have highly restricted, fragmented ranges[17]. They support diverse assemblages of rare or threatened fauna that are also suffering ongoing demographic decline[18].

**Impacts on vascular plant taxa**. We quantified the impact of the Black Summer fires on native vascular plant taxa within the south-east Australian mainland using spatial occurrence data from >9000 species downloaded from the Australasian Virtual Herbarium[19]. Estimates of the proportion of populations or geographic distributions burnt (PF) were developed based on unique specimen location records ($PF_{SR}$), specimen location records binned to a 2.5 x 2.5 km raster grid to reduce oversampling bias ($PF_{BR}$), and for species with ≥10 unique location

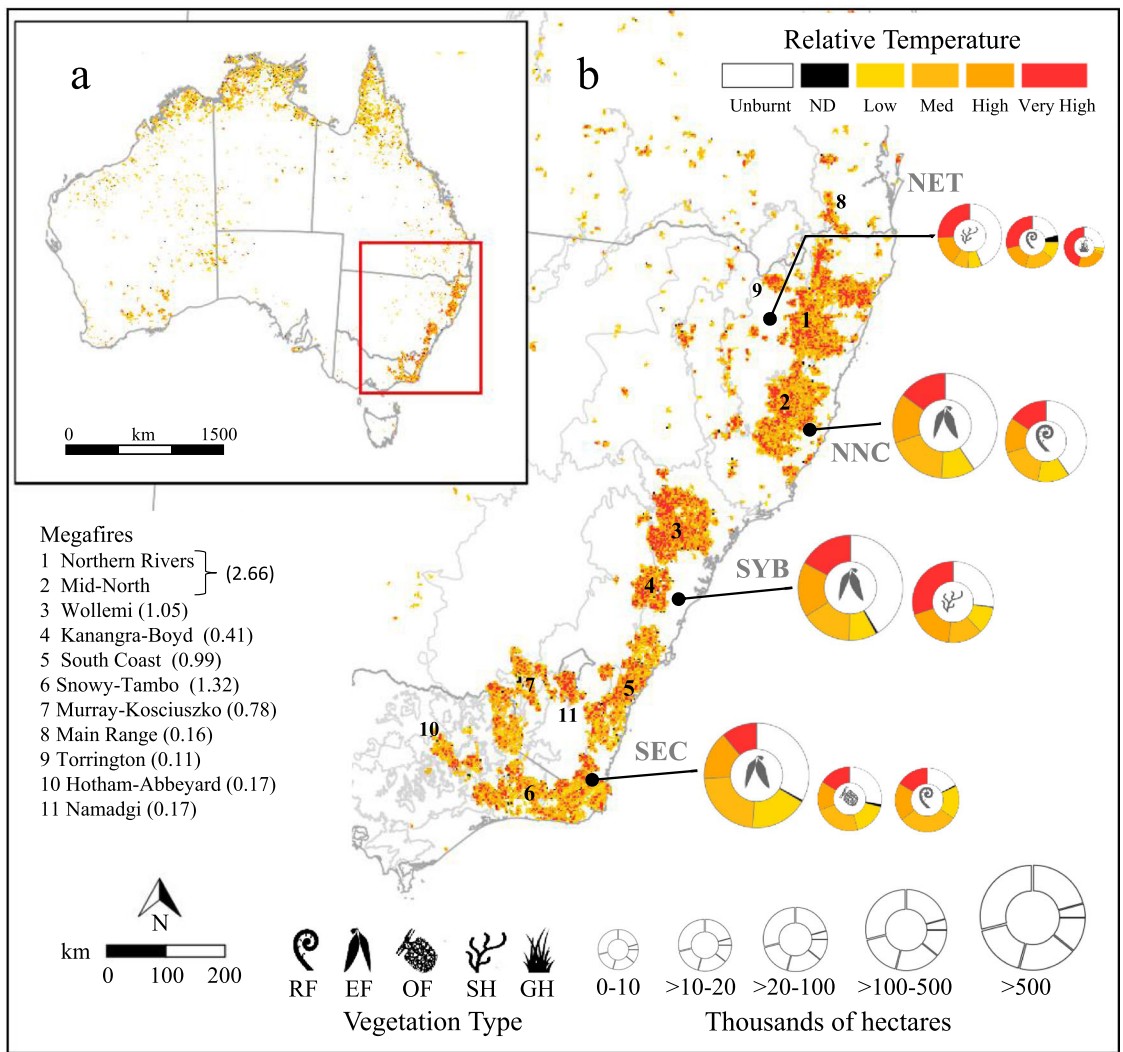

**Fig. 1 Extent and temperature of bushfires between 1 July 2019 and 11 February 2020 and impacts on vegetation. a** Australia, with study area as red inset, and **b** the south-eastern Australian mainland study area, reconstructed from satellite hotspot data (see 'Methods'). Fire temperature was scaled between minimum and maximum temperatures to produce a relative fire temperature ($T_R$; Low = 0–0.25, Med = >0.25–0.50, High = >0.50–0.75, Very High = >0.75–1.0, ND = no data; see 'Methods'). Broad vegetation types are: RF = Rainforest and Vine Thickets; EF = Eucalypt Forests and Woodlands; OF = Other Forests and Woodlands, SH = Shrublands and Heathlands; GH = Grasslands, Herblands, Sedgelands and Rushlands. The total area of vegetation types and proportions burnt in each of four coastal or near-coastal bioregions (NET = New England Tablelands, NNC = NSW North Coast, SYB = Sydney Basin, SEC = South East Corner), and in each relative temperature category are shown. The bracketed value after each megafire is the fire area in millions of hectares.

records, ranges constructed from maximum-entropy (MaxEnt)-based species distribution models ($PF_{SDM}$). We adopted the conservative approach of using the maximum value of all available PF estimates ($PF_M$) as our estimate of fire impact for individual taxa, thus representing an upper limit of the number of taxa impacted (see 'Methods'). We used the criterion of $PF_M >$ 0.50 to identify highly fire-impacted taxa (i.e., >50% of populations or ranges burnt).

Our data indicate that 816 vascular plant species in mainland south-eastern Australia were highly impacted by the Black Summer fires (Supplementary Data 2), of which 325 and 173 were >75% ($PF_M > 0.75$) and >90% burnt ($PF_M > 0.90$), respectively. All known populations of an estimated 116 species (14% of the total) burnt, which is more than double the number of plant species endemic to the British Isles. Among the 816 species, support for $PF_M$ estimates were strong for 80% of taxa ($n = 649$ species) with at least two of $PF_{SR}$, $PF_{BR}$ or $PF_{SDM}$ exceeding

the $PF > 0.50$ criterion for inclusion. The remaining taxa were included solely based on either $PF_{SR}$ ($n = 58$), $PF_{BR}$ ($n = 24$) or $PF_{SDM}$ ($n = 85$); in most cases alternative PF estimates were just below the 0.50 threshold (Supplementary Data 2).

The size of the fires is reflected in both the diversity of plant families ($n = 76$) and life forms affected. Taxa not found in rainforests (RF-) comprised 88% of the flora ($n = 716$ species; Fig. 2e), among which Myrtaceae ($n = 124$ species including 65 *Eucalyptus* spp.), Fabaceae ($n = 104$ including 50 *Acacia* spp.), Proteaceae, Orchidaceae, Asteraceae and Rutaceae were most speciose (43–71 taxa each; Fig. 2e). These taxa occur in a range of habitats, but we estimate that at least 89% are predominantly found either in sclerophyll forest and woodland ($n = 462$) or heathland or shrubland ($n = 119$) or in both ($n = 54$). Most of the remainder occur in swamps and non-woody vegetation (Supplementary Data 2). Among predominantly rainforest, semi-rainforest and rainforest margin species (RF++, RF+ and RF,

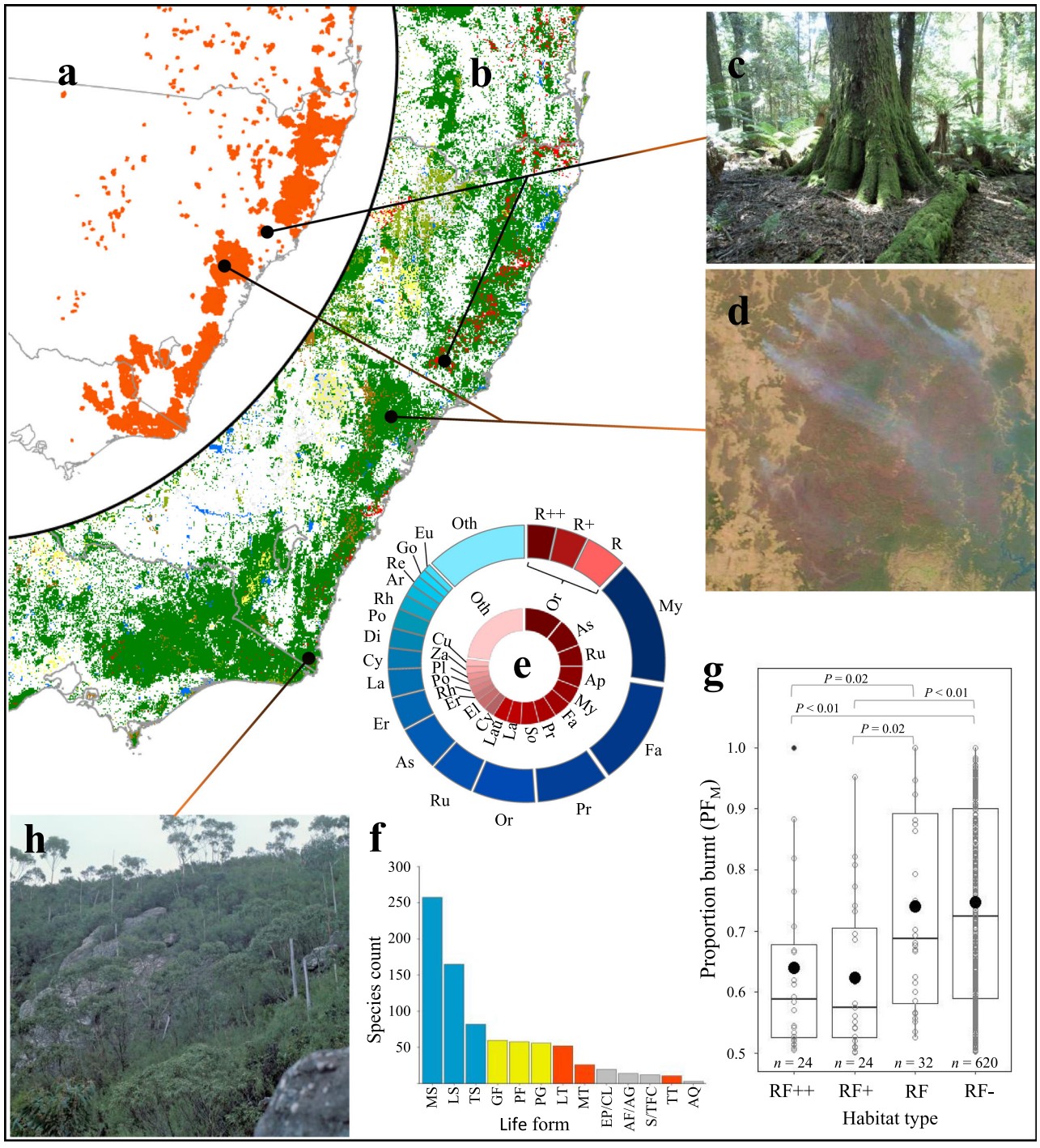

respectively), orchids and members of the Apocynaceae, Solanaceae and Lamiaceae were also diverse (Fig. 2e). The presence of ancient Gondwanan rainforest lineages (e.g., Argophyllaceae, Cunoniaceae, Elaeocarpaceae, Lauraceae, Proteaceae, Trimeniaceae and Winteraceae; Fig. 2e) also support initial fears[20] that the conservation of relict Gondwanan rainforest taxa might be threatened by the Black Summer fires. Notably, however, rainforest taxa (RF++) comprise only $n = 28$ species (3% of the total), of which 24 are endemic to the study area (Fig. 2e). Semi-rainforest and rainforest margin taxa, the majority of which are also found in wet sclerophyll forests, contribute an additional 72 taxa.

The most impacted life forms were dominated by ground- and understory-layer shrubs, with low or prostrate (<1 m tall), medium (1–3 m) and tall forms (>3–7 m tall, including mallee eucalypts), comprising 62% ($n = 504$ species) of all taxa with $PF_M > 0.50$ (Fig. 2f). Of the remainder shorter trees (7–20 m), perennial non-geophytic forbs, geophytes and graminoids each comprised around 7% of the total (Fig. 2f). Most life forms were represented in all major habitat types, apart from annual species, geophytic forbs and taller trees (>35 m), which were almost absent from rainforests, and epiphytes and climbers, which were prevalent in, or entirely restricted to them (Supplementary Table 2). Notwithstanding these minor differences, the fires

**Fig. 2 Landscape-scale implications of the 2019-2020 bushfires. a** Extent of 1 July 2019-11 February 2020 fires. **b** Broad vegetation types in south-eastern Australia; red = Rainforest and Vine Thickets, green = Eucalypt Forest and Woodland, brown = Shrublands and Heathlands, yellow = Grasslands, Herblands, Sedgelands, Rushlands and olive = Other Forests and Woodlands. **c** Cool temperate rainforest at Barrington Tops, NSW. **d** Active fires in the Wollemi megafire on 4 January 2020 with burnt areas and refugia clearly visible. **e** Taxa affected by family and rainforest habitat type (RF + + = rainforest, RF + = semi-rainforest, RF = rainforest margins, all others = non rainforest = RF−). The inner pie chart contains data only for RF + + , RF + and RF species. Ap = Apocynaceae, Ar = Araliaceae, As = Asteraceae, Cu = Cunoniaceae, Cy = Cyperaceae, Di = Dilleniaceae, El = Elaeocarpaceae, Er = Ericaceae, Eu = Euphorbiaceae, Fa = Fabaceae, Go = Goodeniaceae, La = Lamiaceae, Lau = Lauraceae, My = Myrtaceae, Or = Orchidaceae, Pl = Plantaginaceae, Po = Poaceae, Pr = Proteaceae, Re = Restionaceae, Rh = Rhamnaceae, Ru = Rutaceae, So = Solanaceae, Za = Zamiaceae, Oth = other. **f** Number of species by life form type; MS = medium shrubs, LS = low shrubs, TS = tall shrubs, GF = geophytic perennial forbs, PF = other perennial forbs, PG = perennial graminoids, LT = low trees, MT = medium trees, EP/CL = epiphytes and climbers, AF/AG = annual forbs and grasses, S/TFC = short and tall ferns and cycads, TT = tall trees, AQ = aquatic (full descriptions in 'Methods'). Blue = shrubs, orange = trees, yellow = perennial forbs and graminoids, and grey = other groups. **g** Proportion burnt ($PF_M$) of endemic rainforest (RF++), semi-rainforest (RF+), rainforest margin (RF) and non-rainforest (RF-) species. Species counts are shown below data for each habitat type. The middle and lower and upper hinges of each boxplot correspond to the median and first and third quartiles, respectively, while whiskers extend to the largest value no further than 1.5 times the interquartile range. Data points are shown as small circles with outliers filled in black; habitat group means are shown as large black circles. Overall group differences were significant (Kruskal–Wallis $\chi^2$ = 24.5, df = 3, $P < 0.001$); significant pairwise group differences ($P < 0.05$) based on two-tailed Wilcoxon rank-sum tests are shown at the top of the panel. **h** Stand of the rare, endemic mallee-forming *Eucalyptus imlayensis* on Mount Imlay, south coast NSW. Images by (**c**) M.Fagg, (**d**) NASA's Worldview Snapshots application (https://wvs.earthdata.nasa.gov), part of the Earth Observing System Data and Information System, and (**h**) M.Crisp. ©Australian National Botanic Gardens, 1978.

clearly impacted a broad range of species that contribute to both floristic diversity and habitat heterogeneity of forests and woodlands on local to bioregional scales. These characteristics underpin crucial ecosystem services that include biomass production and carbon sequestration[21,22], surface-atmosphere interactions[23] and the provision of foods and habitat for animal assemblages[24–26], and transformational changes in these processes are likely to be of great importance in the wake of the fires.

**Implications for conservation biogeography.** To understand the implications of the Black Summer fires for conservation at broader biogeographical scales[27] we investigated relationships among plant range size, species traits and the location and extent of the Black Summer fires. First, we determined the maximum geographic range extent ($RE_T$) of all vascular plant taxa endemic to the study area and with $PF_M > 50\%$ based on specimen records ($n = 700$ species). These data (Fig. 3a) reveal a strong positive skew (Skewness = 1.66, Shapiro-Wilk normality test $W = 0.803$, $P < 0.001$) towards range-restricted species: 14% ($n = 98$) had extremely ($RE_T < 10$ km) or very small ranges ($RE_T = 10–25$ km) and a further 143 (20%) had ranges of 25–100 km. The presence of a triangular relationship between $RE_T$ and $PF_M$ and in particular a strong inverse linear relationship ($P < 0.001$) at the 0.90 quantile (see Supplementary Notes) shows that, among impacted flora, rare, endemic species were more likely to suffer burning across most or all of their ranges. Indeed, extremely and very range-restricted species experienced fire over an average of 90–95% of their ranges compared with 57–60% for the most widespread species ($RE_T > 750$ km; Fig. 3b; Supplementary Notes). The presence of significant variation in $PF_M$ among 8 increasing $RE_T$ scale classes (Kruskal–Wallis rank-sum test $\chi^2 = 291.4$, df = 7, $P < 0.001$; Fig. 3b) further supports this conclusion.

Analysis of variation in $RE_T$ also indicate significant differences in range extent across life form categories (Fig. 3c; Kruskal–Wallis rank-sum test $\chi^2 = 32.2$, df = 11, $P < 0.001$), with epiphytes and climbers and perennial graminoids tending to have larger ranges than shrubs, low trees and geophytic forbs. The relationship between grouped $RE_T$ size classes and life form categories (Supplementary Table 3 and Supplementary Notes) also shows strong contingency ($\chi^2 = 42.1$, df = 15, $P < 0.001$) between these variables, with medium and low shrubs more likely to have narrow endemic ranges (<25 km) and/or less likely to have moderate (100–500 km) to large ranges (>500 km; Fig. 3c and Supplementary Notes). These data are broadly consistent

with prior studies[28] that show a prevalence of species with small and threatened ranges among shrub-rich eastern Australian plant families (e.g., Myrtaceae, Fabaceae, Lamiaceae, Proteaceae, Rutaceae and Ericaceae).

Mapping of burnt populations of endemic taxa with very small to extremely small ranges ($RE_T < 25$ km) shows that most were concentrated in a small number of impact hotspots from far southern Queensland to south-east NSW, notably in the Border Ranges, Washpool-Gibraltar region of northern NSW, the Wollemi-Blue Mountains area west of Sydney and isolated ranges in southern NSW (Fig. 4a). Remaining unburnt refugia for these taxa are now small and disjunct, the most species-rich of which appears to lie inside Blue Mountains National Park (Fig. 4a). Range-restricted taxa ($RE_T = 25–100$ km) were similarly affected but had more numerous and larger impact hotspots and refugia (Supplementary Fig. 3a). These patterns clearly demonstrate the emergent biodiversity implications of megafires that collectively exceed the size of local species' ranges, span multiple bioregions, and fall within areas of high plant diversity and endemism. In contrast, the large number and extent of unburnt refugia for very widespread (Fig. 4b) and moderately widespread taxa (Supplementary Fig. 3b) despite the size of associated burnt areas reflects their generally smaller $PF_M$ values (50–75%, Fig. 3b) and indicates a higher level of resilience to fire events of this scale.

Rainforest taxa were burnt in five main north-eastern NSW richness hotspots (Fig. 4c), three of which lie inside the 2.66 Mha Northern Rivers-Mid North megafire complex (Fig. 1b). Concentrations of burnt taxa occurred within the Main Range, Nightcap Range, and Gibraltar-Washpool regions, which together comprise a vital part of the World Heritage listed Gondwanan Rainforests of Australia[29]. Unburnt refugia containing these species now primarily occur in the Springbrook-Lamington-Mt Jerusalem-Nightcap region, some of New England National Park and the Barrington Tops (Fig. 4c). Notably rich areas of burnt semi-rainforest and rainforest margin species occurred further south in NSW (Fig. 4d). While rainforest and semi-rainforest species appeared to have generally larger range extents than other taxa (analysis of variance $F = 4.7$; df = 3, 696; $P = 0.003$; mean $RE_T$ of RF+ and RF- species differ at the $P < 0.01$ level, see Supplementary Notes) and had a significantly smaller proportion of these burn (Kruskal–Wallis rank-sum test $\chi^2 = 24.5$, df = 3, $P < 0.001$; Fig. 2g), the small size and disjunct distribution of these refugia is concerning, and their protection and management may now be a priority. Species-rich unburnt refugia for the more

**Table 1 Impact of the 2019–2020 bushfires on vegetation types within the study area. Shown are area burned (FG$_i$; thousands of hectares) and as a percentage of the total (PF$_T$) across all 11 bioregions and in four coastal and near-coastal bioregions. Only Major Vegetation Groups >50% burnt (PF$_T$ > 50%) and with an impact score (I$_H$) > 0.25 (see 'Methods') in at least one individual bioregion are shown.**

| Broad vegetation type[a] / Grouped bioregions | | All bioregions[b] | | Coastal bioregions[c] | | Individual bioregions[d] | | | | | | | |
|---|---|---|---|---|---|---|---|---|---|---|---|---|---|
| Code | Major vegetation group (MVG) | FG$_i$ | PF$_T$ | FG$_i$ | PF$_T$ | AUA FG$_i$ | NAN FG$_i$ | NET FG$_i$ | NNC FG$_i$ | SEC FG$_i$ | SEH FG$_i$ | SEQ FG$_i$ | SYB FG$_i$ |
| **Rainforests and Vine Thickets** | | | | | | | | | | | | | |
| 1 | Rainforests and Vine Thickets | 328.90 | 33.0 | 244.65 | 58.6 | | | 13.06[C] | 175.97[D] | 32.70[D] | | | |
| **Eucalypt Forest and Woodlands** | | | | | | | | | | | | | |
| 2 | Eucalypt Tall Open Forests | 1357.33 | 53.0 | 1116.97 | 56.0 | | 11.40[B] | 144.24[C] | 753.62[C] | 152.76[D] | 80.42[C] | | 700.48[C] |
| 3 | Eucalypt Open Forests | 3849.54 | 33.6 | 2246.06 | 64.4 | | | | 479.76[C] | 816.58[D] | | | |
| 4 | Eucalypt Low Open Forests | 102.33 | 56.7 | 84.78 | 58.4 | | | 18.89[C] | 65.04[D] | 0.57[E] | 1.05[D] | 9.55[D] | |
| 5 | Eucalypt Woodlands | 1899.54 | 17.9 | 904.31 | 52.3 | | | | | 389.82[D] | | | |
| 11 | Eucalypt Open Woodlands | 129.14 | 10.9 | 44.89 | 36.4 | | | | 0.03[A] | | 11.52[D] | | 33.00[C] |
| | Total | 7337.88 | 28.2 | 4397.00 | 58.8 | | | | 1,323.74[C] | 1,360.56[D] | | | 1,245.70[C] |
| **Other Forests and Woodlands** | | | | | | | | | | | | | |
| 6 | Acacia Forests and Woodlands | 41.24 | 4.1 | 12.39 | 68.9 | 0.013[C] | | | | 12.32[D] | 2.15[E] | | 0.01[B] |
| 9 | Melaleuca Forests and Woodlands | 33.84 | 25.2 | 7.20 | 29.2 | | | | | | 0.17[E] | | |
| 10 | Other Forests and Woodlands | 38.19 | 53.8 | 33.40 | 70.7 | | | | | 33.07[D] | | | |
| 13 | Acacia Open Woodlands | 0.04 | 0.6 | 0.04 | 100 | | | 0.04[A] | | | | | |
| 15 | Low Closed Forests and Tall Closed Shrublands | 14.10 | 13.5 | 0.14 | 27.7 | | | | | | 0.55[D] | | |
| | Total | 248.96 | 8.5 | 100.85 | 38.4 | | | | | 46.56[D] | | | |
| **Shrublands and Heathlands** | | | | | | | | | | | | | |
| 14 | Mallee Woodlands and Shrublands | 19.53 | 7.5 | 12.81 | 79.4 | | | 4.27[A] | | | | | 6.99[B] |
| 16 | Acacia Shrublands | 2.87 | 13.8 | 1.34 | 24.8 | | | | 0.20[B] | | | | |
| 17 | Other Shrublands | 20.20 | 12.7 | 11.54 | 31.1 | | | | | | | | 1.93[B] |
| 18 | Heathlands | 175.1 | 55.6 | 148.90 | 68.1 | | | 3.64[A] | | 12.30[D] | 10.59[B] | | 128.08[B] |
| | Total | 217.74 | 28.9 | 174.59 | 63.0 | | | 11.95[D] | | | | | 137.04[B] |
| **Grasslands, Herblands, Sedgelands, Rushlands** | | | | | | | | | | | | | |
| 19 | Tussock Grasslands | 26.38 | 3.4 | 0.93 | 14.0 | | | 0.06[C] | | | | | |
| 21 | Other Grasslands, Herblands, Sedgelands and Rushlands | 32.5 | 20.3 | 11.22 | 42.0 | | | 5.25[A] | | | | | |
| | Total | 59.12 | 6.24 | 12.4 | 36.2 | | | 5.31[A] | | | | | |

[a]NVIS Major Vegetation Groups (names and NVIS codes are shown for each) were grouped into four broad vegetation types. The full list of MVGs in each broad vegetation type is provided in Supplementary Table 1.
[b]Bioregions included in the study were: Australian Alps (AUA), Brigalow Belt South (BBS), Nandewar (NAN), New England Tablelands (NET), NSW North Coast (NNC), NSW South Western Slopes (NSS), South East Coastal Plain (SCP), South East Corner (SEC), South Eastern Highlands (SEH), Southeast Queensland (SEQ) and Sydney Basin (SYB).
[c]NET, NNC, SYB, SEC.
[d]Fire area data provided only for severely impacted vegetation types with PF$_T$ > 0.50 and impact score I$_H$ > 0.25 in at least one bioregion (see 'Methods'). Letters refer to I$_H$ categories: A = I$_H$ > 80%, B = I$_H$ > 65–80%, C = I$_H$ > 50–65%, D = I$_H$ > 35–50% and E = I$_H$ > 25–35%.

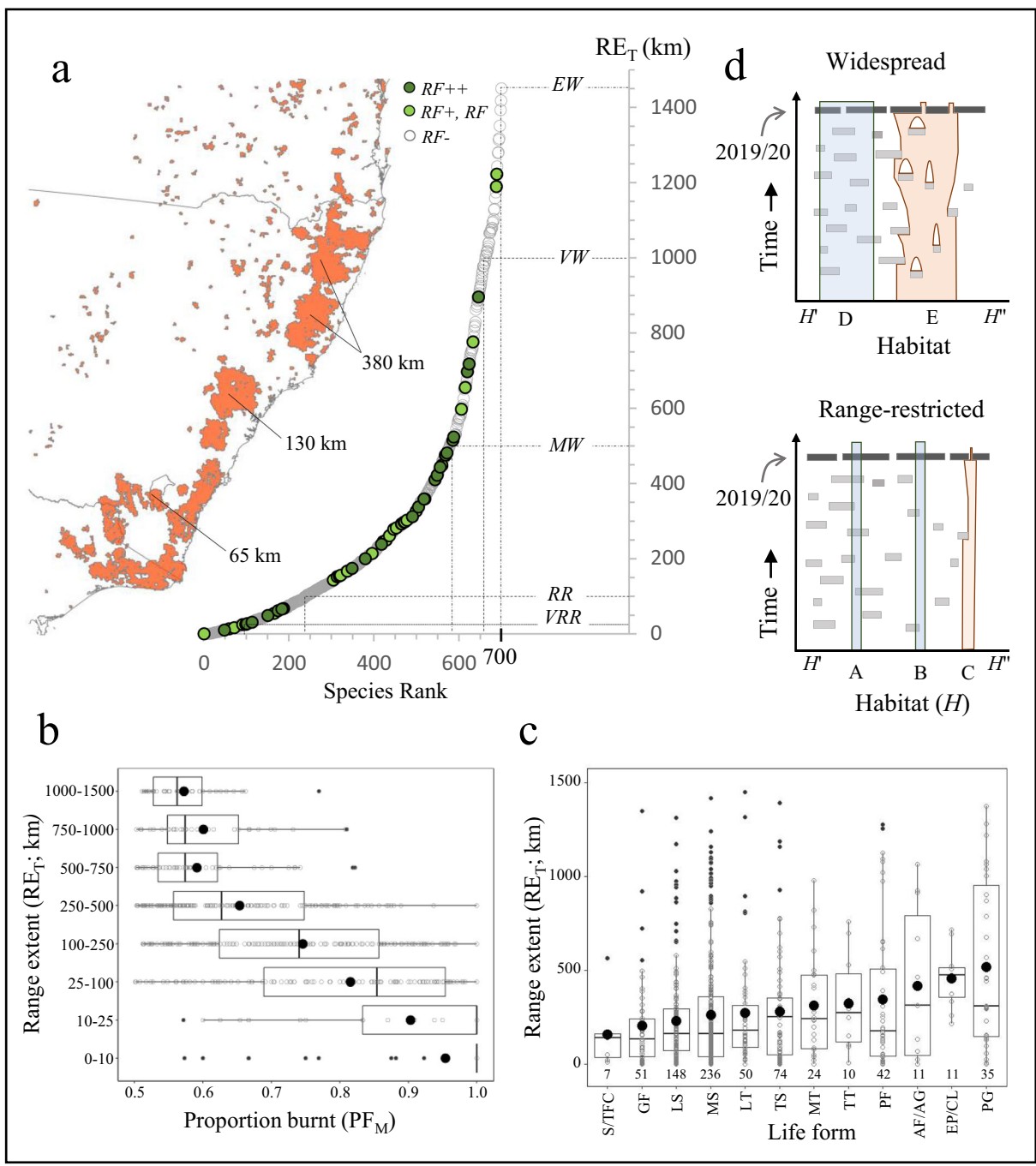

diverse non-rainforest flora were comparatively numerous and extensive (Supplementary Fig. 3c).

Despite the immediate potential impacts on south-eastern Australian vegetation revealed in this study, the ability of many plant communities and species to recover and regenerate after megafires of this scale remain poorly understood[30–32]. As we have shown, the size of species ranges (Fig. 3a, b) and the geographic position of the fires (Figs. 2a and 4a–d) both played an important role in determining the diversity and composition of the fire-affected flora. The demographic impact of the fires on specific taxa will also depend on their ability to survive and recover from fire (i.e., fire persisters vs. non-persisters sensu Pausas et al.[33]). Figure 3d contains a simple spatio-temporal framework that integrates these concepts, in which range-restricted and wide-spread fire persister and non-persister species occur in habitats with different fire histories. Here, we use this framework to

investigate the implications of the fires for five different types of taxa.

Our data show that the majority of species affected by the fires are primarily found in sclerophyll forests and woodlands or shrublands and heathlands (Supplementary Data 2 and Supplementary Fig. 3d). Fire is a natural part of these ecosystems and many species are highly fire-adapted with traits such as a soil-stored seed bank[34], serotinous cones or fruits[35], smoke- and/or heat-induced seed germination[36,37], fire-cued flowering[38], thick protective basal bark[39], epicormic buds or underground lignotubers that either provide protection from fire and/or ensure subsequent recovery[40]. Evidence from 270 species in our study confirms this pattern: 251 (93%) across 93 genera are reportedly fire persisters that can resprout or regenerate via propagules after fire, or both (Fig. 4e and Supplementary Data 3). Among these are many rare, endemic taxa such as some mallee eucalypts (Fig. 2h),

**Fig. 3 Geographic ranges and characteristics of vascular plant taxa burnt (PF$_M$ > 0.50) during the 2019–2020 bushfires. a** Taxa endemic to the study area ranked by maximum range extent (maximum distance between specimen locations; RE$_T$; y-axis). Upper RE$_T$ limits and species counts for very range-restricted (<25 km, VRR), range-restricted (25–100 km, RR), moderately widespread (100–500 km, MW), very widespread (500–1000 km, VW) and extremely widespread (1000–1500 km, EW) are shown. The approximate maximum extent of three megafires ranging from small (Namadgi megafire, 0.17 Mha) medium (Wollemi megafire, 1.1 Mha) and very large (Northern Rivers-Mid North complex; 2.66 Mha are shown (c.f., Fig. 1). **b** Relationship between range size (RE$_T$) and proportion burnt (PF$_M$) for endemic taxa, showing an increase in PF$_M$ with declining range size. Differences among range size categories were significant (Kruskal–Wallis rank-sum test $\chi^2 = 291.4$, df = 7, $P < 0.001$) based on $n = 700$ species across eight range size classes. For each boxplot the middle and lower and upper hinges correspond to median and first and third quartiles, respectively, while whiskers extend to the largest value ≤1.5 times the interquartile range. Data points are shown as small circles with outliers filled in black; group means are shown as large filled circles. **c** Variation in range extent with life form of endemic species (excluding aquatic species; $n = 1$). Life form acronyms are as in Fig. 2f. Boxplots were constructed as in Fig. 3b; species counts are shown below each group. Differences among life forms were significant (Kruskal–Wallis rank-sum test $\chi^2 = 32.2$, df = 11, $P < 0.001$) based on $n = 699$ species across 12 life form classes. Significant group differences determined using two tailed pairwise Wilcox rank-sum tests were: PG vs. GF, LS and MS (all $0.01 > P > 0.001$) and EP/CL vs. GF, LS ($0.01 > P > 0.001$), MS ($P = 0.016$), LT ($P = 0.018$) and TS ($P = 0.049$). No other differences were significant at the 0.05 criterion. **d** Conceptual framework for species responses to the Black Summer fires based on range size (widespread vs. range-restricted), fire persistence (non-persister = red, persister = blue) and habitat (H′ = frequent history of fire, H″ = infrequent or null history of fire). Type A and D species have small and large ranges, respectively, and occur in fire prone landscapes; Types B, C and E occur in less fire-prone landscapes that were heavily burnt during 2019–2020. The Black Summer fires were likely unprecedented for Types B–E.

shrubby *Acacia*, *Callistemon*, *Grevillea* and *Zieria*, and the tuber-forming orchids *Corunastylis* and *Paraprasophyllum*. Despite having small, fire-prone ranges, such fire-persister taxa (Type A in Fig. 3d) appear to have generally recovered well from other recent bushfires in south-eastern Australia[41], and for these the Black Summer fires are unlikely to represent an unprecedented event, unless they have undergone recent range contraction. Many also occur in rocky habitats (cliffs, granite tors etc., Supplementary Fig. 3e) or in gullies or gorges (Supplementary Data 2) that afford some protection from fire.

In contrast, for widespread endemic species with ranges of 500 km or more ($n = 122$; Fig. 3a) the demographic consequences of the 2019-2020 fires are likely unprecedented over at least the past two centuries. While the majority are also likely to be fire-persisters (Type D in Fig. 3d and Supplementary Data 3), they are now at risk of novel range-wide threats during the recovery phase such as dieback and inhibited post-fire recovery caused by myrtle rust (*Austropuccinia psidii*)[42], herbivory of regrowth by invasive animals, and drought. Rainforest taxa capable of surviving fire but unable to compete with subsequent incursion of weeds or sclerophyllous species (Type B in Fig. 3d) may be under similar pressure[43,44]. Obligate seeding woody species such as the ash eucalypts (e.g., *Eucalyptus fraxinoides*) are likely to be under threat if fires return prior to completion of their typically long sexual maturation periods[45,46].

Finally, plants with narrow or wide ranges that rely on recolonisation via dispersal of propagules from unburnt areas (fire non-persisters; Types C and E; Fig. 3d) are clearly at risk of demographic decline and range contraction[47,48]. Among those identified (Supplementary Data 3) most were geophytic orchids with specialised structures that are damaged by fire (e.g., shallow tubers in *Caladenia*[49] and *Chiloglottis*), wind-dispersed shrubs, or obligate epiphytes that lack a seedbank (e.g., the orchids *Dockrillia* and *Plectorrhiza*). Notable impact richness hotspots for these taxa occurred in north-eastern rainforests, the southern ranges, and the far south-east corner of NSW (Fig. 4f), and while no species endemic to the study area had very small ranges (RE$_T$ < 25 km), populations of some epiphytes now apparently occur in highly disjunct rainforest patches (Supplementary Fig. 3f). A small but evolutionarily significant number of fire sensitive Gondwanan rainforest relics[50,51] and other rainforest species may also face decline, particularly in habitats, which rarely, if ever, burn. Some of these taxa are known to resprout[52] or regenerate from seed[53] but for most further investigation of these traits is urgently required.

Collectively, there are grounds for cautious optimism that most plant species identified here will recover from all but the most intense fire[54–58]. Despite this resilience, however, recent evidence from forested ecosystems globally suggest that catastrophic fire events are increasingly catalysing dramatic changes in species composition across large areas[59,60]. In the most extreme cases tipping points are being reached, resulting in transitions from forest to non-forested vegetation[61]. Impairment of post-fire regeneration has been specifically linked to thresholds in vapour pressure deficit, soil moisture and maximum surface temperature[1,31], as well as fire intensity and seed availability[1,62]. This is particularly concerning because much of the vegetation affected by the Black Summer fires was already suffering from extreme drought, record high temperatures (Supplementary Figure 1) and patchy canopy dieback prior to the onset of the 2019-2020 fire season. Even in the absence of fire these factors can drive rapid shifts in the dynamics and distribution of forest ecosystems[2,63,64]. In regions where the Black Summer fires burnt areas that had only recently recovered from previous fires, increasing fire frequency will be an additional stressor[58]. Australia's Gondwanan rainforest communities (Fig. 2c), which were severely burnt in several bioregions (Fig. 1 and Table 1), are probably most susceptible. Furthermore, our data indicates that the vast scale of the 2019-2020 bushfires caused a taxonomically diverse array of species with subcontinental-scale ranges to suffer extensive losses of mature individuals, rendering them potentially susceptible to other factors such as disease[65], herbivory, and disturbance. Collectively, these factors are all likely to have depleted the resilience of some forested ecosystems to fires of the severity and magnitude as those experienced during the 'Black Summer'. Further work is now required to determine whether they may now be undergoing regenerative failure and permanent biogeographic change.

## Methods

**Generation of fire maps from hotspot data.** Fire maps were derived from hotspot data obtained from the Geoscience Australia - Digital Earth Australia website (https://hotspots.dea.ga.gov.au/) for the period 1 July 2019 to 11 February 2020. We used data from four satellites: Himawari 8 [sensor = AHI, process algorithm = WFABBA version 6_5_010g[66], hotspot temperature (T; Kelvin)], Suomi NPP [(sensor = VIIRS, process algorithm = VCM 1.O.000.002, hotspot confidence (%)], Aqua [sensor = MODIS, process algorithm = MOD14 version 6.2.1[67], hotspot temperature ($T_M$; K) and confidence (%)], and Terra [sensor = MODIS, process algorithm = MOD14 version 6.2.1, hotspot temperature (K) and confidence (%)]. This dataset contained >3 million geocoded hotspot entries. The data were first cleaned to include only values of $T > 0$ K (Aqua, Terra, Himawari 8). We then removed spurious hotpots that did not occur within known fires by selecting

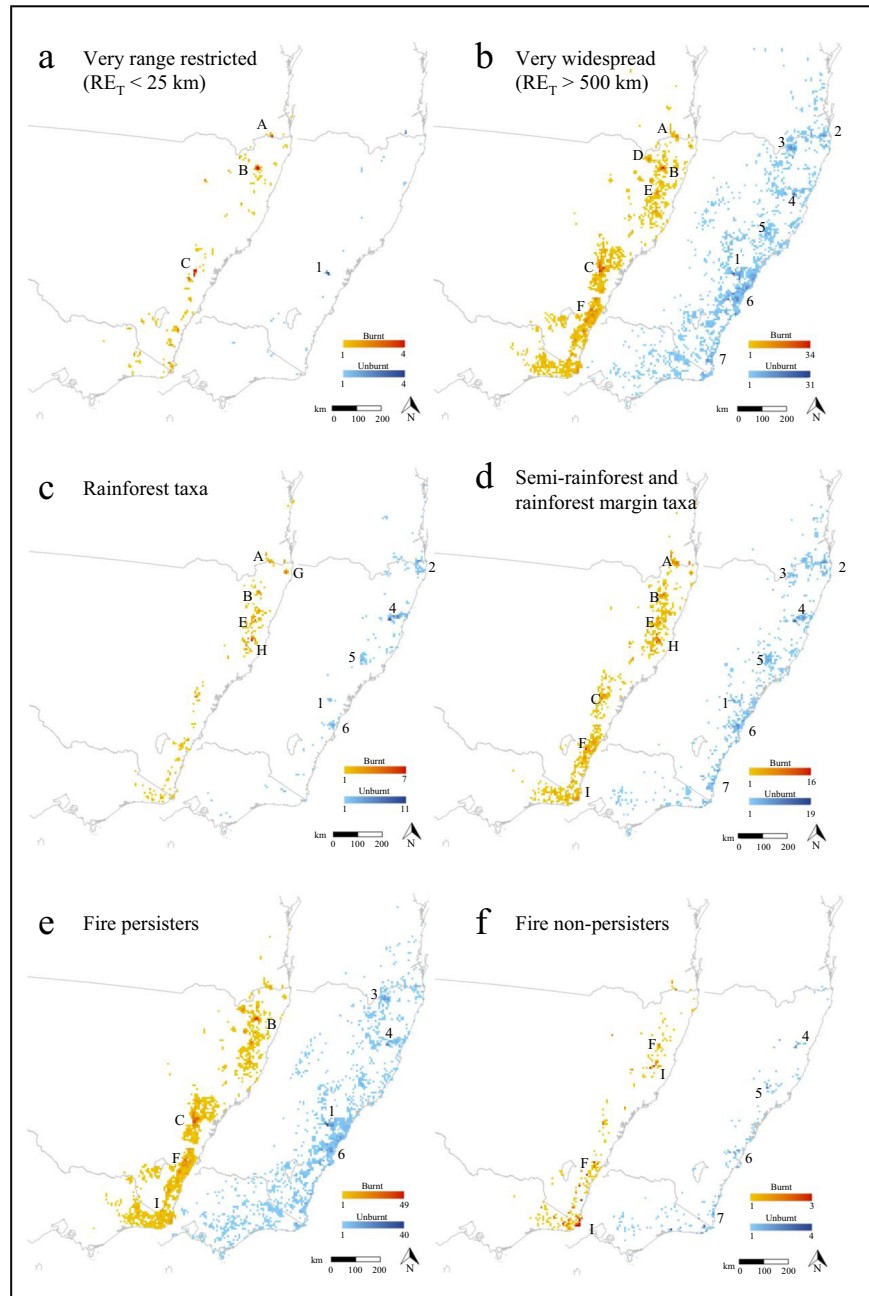

**Fig. 4 Species richness based on burnt (red) and unburnt (blue) location records of species with >50% of populations or ranges burnt aggregated to a 7.5 × 7.5 km grid.** Shown are: burnt and unburnt maps for very range-restricted species (**a**; maximum range extent ($RE_T$) < 25 km; includes extremely range-restricted species with $RE_T$ < 10 km), very widespread species (**b**; $RE_T$ > 500 km; includes extremely widespread species with $RE_T$ > 1000 km), rainforest taxa (**c**; RF++ species), semi-rainforest (RF+) and rainforest margin (RF) taxa (**d**), species that persist after fire (**e**) and species that do not persist after fire (**f**). Burnt species richness hotspots are: A = Border Ranges, B = Washpool-Gibraltar Range, C = Wollemi – Blue Mountains, D = Torrington, E = Guy Fawkes, F = Budawang, G = Nightcap Range, H = Kumbatine-Willi Willi, I = South East Corner. Unburnt refugia are: 1 = Blue Mountains National Park, 2 = Springbrook–Lamington–Mt Jerusalem–Nightcap, 3 = Sundown-Donnybrook, 4 = New England National Park east of Bellingen, 5 = Barrington Tops, 6 = Sydney-Budderoo-Macquarie Pass, 7 = South East Corner.

only hotspots with $T > 500$ K (Himawari 8; 2,627,414 records) or hotspot confidence >50% (Aqua and Terra (209,224 records) and Suomi NPP (151,801).

The data were then gridded using a 2.5 × 2.5 km grid and the maximum T ($T_M$) and/or confidence value determined for data from each satellite for each grid cell. A single raster containing all cells with identified hotspots was then generated. Contiguous cells containing at least one hotspot were then polygonised using the r Raster package, and polygons ≤25 km² and lacking any cells with $T > 1000$ K (Himawari 8) were removed from the dataset. The retained dataset, therefore, contained only large fires with at least one hotspot indicating a moderate (1000–1500 K) emission temperature or higher. Finally, to gain insight into the temperature of fires within each grid cell we re-scaled all observed hotspot

temperatures from the Himawari 8 satellite, or if absent, from Aqua and Terra satellites, to a relative scale of between 0 and 1, which represented maximum and minimum values of $T$ [302.1 K to 506 K (29–233 °C) for Aqua and Terra and 400 to 1999.9 K (127–2273 °C) for Himawari 8 data] across the entire dataset (termed the relative fire temperature, $T_R$). We then classified $T_R$ into four relative fire temperature classes: Low = 0–0.25, Medium =>0.25–0.50, High = >0.50–0.75, Very High = >0.75–1.0, with an additional class (no data) for cells containing >50% fire confidence, but no fire temperature data. The final fire maps included all internal unburnt polygons (minimum one grid cell). Data manipulation was performed using the R programming environment Version 4.0.1 [R Core Team (2013). R: A language and environment for statistical computing. R Foundation for

Statistical Computing, Vienna, Austria]. Software packages used here and below are documented in the Supplementary Notes.

The data filtering decisions removed spurious cells containing hotspots that lay outside of known fires and had little or no impact on the spatial arrangement of any of the major fires that were the focus of the study. The modelled fire extent was consistent with maps provided by the Emergency Management of Spatial Information Network Australia (www.emsina.org). However, it is important to point out that due to the resolution of the gridded dataset ($2.5 \times 2.5$ km) the estimated fire area may exceed the extent of the actual burnt area, particularly around the fire margins, and that the relative fire temperature is the maximum observed for any hotspot data point within a given grid cell based on one of two process algorithms. Further refinement of these data may alter associated fire maps. Hotspot data may also underestimate fire occurrence during cloudy weather or other adverse conditions.

**Quantification of fire impacts on vascular plant species**. The objective of our study was to quantify the impact of the 2019–2020 fires on plant species and communities occurring in the south-east Australian mainland. The core metric used to compare species-and community-level impacts is the proportion of total species records or estimated area that fell within the modelled 2019–2020 fires, here defined as $PF = FG_I/(FG_I + FG_O)$ where $FG_I$ and $FG_O$ are total number or records or area inside and outside of the burnt areas, respectively. For all plant community data (major vegetation groups or MVGs; see below) PF was simply estimated as the proportion or percentage of each MVG (or combination of MVGs) or bioregion(s) that occurred within the fires ($PF_T$). We used three different approaches, including species distribution modelling, to estimate PF for individual plant species based on specimen collections held at herbaria across Australia, because location data and ranges constructed from herbarium specimen data suffer from known spatial biases[68]. The underlying datasets and quantitative approaches used to estimate PF are described below.

**Bioregion and vegetation mapping**. We determined the areal proportion of each of eleven major bioregions in the south-eastern study area that were inside the 2019–2020 fires ($PF_T$) based on bioregions defined by the Interim Biogeographic Regionalisation for Australia (IBRA)[8] available at https://www.environment.gov.au/land/nrs/science/ibra and downloaded from https://www.environment.gov.au/fed/catalog/search/resource/details.page?uuid=%7B4A2321F0-DD57-454E-BE34-6FD4BDE64703%7D. Within each bioregion we also determined the area of each Native Vegetation Information System (NVIS) Major Vegetation Group (MVG)[7] based on data managed by the Australian Government Department of Agriculture, Water and the Environment available at http://www.environment.gov.au/land/native-vegetation/national-vegetation-information-system/data-products and downloaded from http://environment.gov.au/fed/catalog/search/resource/details.page?uuid=%7B991C36C0-3FEA-4469-8C30-BB56CC2C7772%7D. To allow for comparisons to be made across general vegetation types that vary in species composition but retain a similar physiognomic structure, we also grouped 22 MVGs into four broader vegetation classes. These were: (1) rainforests and vine thickets (MVG 1), (2) eucalypt forests and woodlands (MVG 2-5, 11), (3) other forests and woodlands not typically dominated by eucalypts (MVG 6–10, 13, 15, 31), (4) shrublands and heathlands, including some open mallee woodlands (MVG 14, 16–18), and (5) grasslands, herblands, sedgelands and rushlands (19, 20, 21, 22; Supplementary Table 1). We then determined $PF_T$ for each MVG and broad vegetation class that occurred within the 2019–2020 fires, across (1) all bioregions, (2) across four coastal and near-coastal bioregions that were most affected by the fires (New England Tablelands, NSW North Coast, South East Corner, and Sydney Basin), and (3) within each bioregion. For each MVG with $PF_T > 50\%$, we also determined, at the bioregional level, an impact metric, $I_H$, the estimated percentage of area in very high and high relative fire temperature classes relative to the total fires area, such that $I_H = [(A_{VH} + A_H)/(A_{TOT})] \times 100$ where $A_{VH}$ = total area burned at very high relative hotspot temperature ($T_R > 0.75$–1.0), $A_H$ = total area burned at high relative hotspot temperature ($T_R > 0.50$–0.75) and $A_{TOT}$ is the total area burned (excluding cells lacking temperature data).

**Herbarium specimen dataset**. Herbarium specimen records for south-eastern Australia [extent = bottom left (143.7012, −43.94065) to top right (156.58, −23.1593)] were downloaded from the Australasian Virtual Herbarium on January 24, 2020 (https://doi.org/10.26197/5e2bcb71d290c). We restricted the dataset to vascular plant species (Phylum Tracheophyta, included now Phylum Charophyta in the AVH), a total of more than 1.4 million records. We then screened all records for the presence of cultivated plants, introduced species, collections made prior to 1950 and specimens with questionable taxonomic attribution (including hybrids) or incorrect or poor quality (spatial resolution >25,000 m) location data and these were removed to improve data quality. We checked data for the presence of synonyms and outdated species names, taxonomic errors and recent taxonomic revisions and corrected the database. For all species with incomplete taxonomic information we determined, and added, plant family. Taxonomy and nomenclature followed the Australian Plant Census (APC; https://www.anbg.gov.au/chah/apc/) and for species not listed in the APC the Australian Plant Name Index (https://www.anbg.gov.au/apni/).

We also removed all species-level spatial duplicates to reduce bias associated with the oversampling of plants at a small number of sites within larger species ranges. The data were then cropped to the final southeast mainland study area of 144.01, −39.17 (bottom left) to 154.0, −25.34077 (top right), leaving a total of ~700,000 specimen records for use in estimating fire impacts. This region was selected because it encompassed all megafires that occurred in southeast Australia during the 2019–2020 bushfire season. Our data therefore explicitly include only specimen records from inside of the south-east Australian study area and so estimates of species-wide fire impacts pertain only to the parts of species ranges that occur within this area. Our data indicate that 86% of taxa with $PF_M > 0.50$ were endemic to the study area and another 4% nearly so (see below and Supplementary Data 2).

Screening and manipulation of the database was performed using R version 4.0.1 and OpenRefine version 3.3 (Copyright 2010–2012, Google Inc. and contributors; https://openrefine.org/).

**Estimation of fire impacts and species ranges based on herbarium specimen data and species distribution models**. We estimated PF for all species in the cleaned herbarium specimen record database using two direct methods. First, we estimated the fraction of each spatially unique specimen location records that fell within the modelled fires ($PF_{SR}$) by overlaying species-level data with the gridded fire layer. The metric $PF_{SR}$ assumes that the spatial distribution of records is an unbiased sample of the underlying species distribution, which may underestimate the realised niche of plant species. However, it may be more accurate than predictive models for rare species that have been targeted for intensive collection and for which ranges have been well established. Second, to reduce the impact of spatial oversampling bias along roadsides and other heavily travelled areas[68] we binned the same specimen location records into a $2.5 \times 2.5$ km spatial grid and determined, for each species, the fraction of cells that fell within the fires ($PF_{BR}$). Since $PF_{SR}$ and $PF_{BR}$ are based on known location data, neither account for the possible presence of populations in unsampled areas. We also determined the maximum verified geographic range extent ($RE_T$; km) of each species as the difference between the two most physically distant location records within the study region[69]. We then classified $RE_T$ into six size classes: (1) extremely range restricted (<10 km), very range-restricted (10–25 km), range-restricted (25–100 km), moderately widespread (100–500 km), very widespread (500–1000 km) and extremely widespread (>1000 km).

We constructed potential species distributions using the predictive maximum-entropy (MaxEnt) method based on the environmental conditions of known (positive) specimen locations extracted from the herbarium specimen record database. Models were considered only for species containing 10 or more unique geographic locations, which is towards the extreme lower sample size at which MaxEnt models are likely to produce reliable results[70–72]. MaxEnt species distribution models were constructed using MaxEnt version 3.4.1 (available at https://biodiversityinformatics.amnh.org/open_source/maxent/) and the R package dismo version 1.3–3 using all specimen records as training data, 21 environmental variables as predictor layers at 1 km² resolution, and 10,000 background points.

Environmental predictors for the MaxEnt models included 15 bioclimatic variables and six soil and landform variables. Bioclimatic variables were temperature seasonality (CV; Bio04), maximum temperature of warmest month (°C; Bio05), minimum temperature of coldest month (°C; Bio06), mean temperature of warmest quarter (°C; Bio10), mean temperature of coldest quarter (°C; Bio11), annual precipitation (mm; Bio12), precipitation seasonality (CV; Bio15), precipitation of driest quarter (mm; Bio17), annual mean moisture index (Bio28), moisture index seasonality (CV; Bio31), mean moisture index of wettest quarter (Bio32), annual total actual evapotranspiration (mm; EAA), minimum monthly potential evaporation (mm; EPI), minimum monthly atmospheric water deficit (mm; WDI) and maximum monthly atmospheric water deficit (mm; WDX). The six soil and landform variables included soil clay content 0–30 cm (%; CLY), sand content 0–30 cm (%; SND), total soil nitrogen 0–30 cm (%; NTO), total soil phosphorus 0–30 cm (%; PTO), soil bulk density 0–30 cm (g/cm³; BDW) and topographic wetness index (TWI3S). Data (9 s resolution) are available from the CSIRO Data Access Portal at https://doi.org/10.25919/5dce30cad79a8 (Bio4-Bio32)[73], https://doi.org/10.4225/08/5afa9f7d1a552 (EAA, EPI, WDI and WDX)[74] and https://doi.org/10.4225/08/5b285fd14991f (CLY, SND, NTO, PTO, BDW and TWI3S)[75].

For each species model the predicted probability of occurrence was determined using the complementary log–log (cloglog) link function[76]. We used two different thresholds to generate final binary predictive distribution maps: (1) the maximum of Cohen's Kappa (K)[77–79], which determines the optimal threshold for statistical discrimination of presence-absence, and (2) the maximum training sensitivity plus specificity (MTSS), which is equivalent to the finding the point in the receiver operator characteristic (ROC) curve with a tangent slope of 1[80]. We also determined the tenth percentile training presence (P10) threshold, which allows a fixed presence omission rate of 10%, for use in validation of MTSS models. All models were then overlayed with the fire layer, producing estimates of the fraction of the estimated range that fell into the fires based on K, MTSS and P10 thresholds ($PF_K$, $PF_{MTSS}$ and $PF_{10}$). While K-based models tended to accurately reproduce the small distributions of severely range-restricted species, they frequently underestimated the ranges of species with large but very disjunct or clumpy distributions and hence had excessively high positive training omission rates. We therefore excluded all K-based models with a true-positive rate (TPR, the

proportion of known positive records correctly classified) below 0.80. We also found that a small number of MTSS-based models overestimated the distributions of very rare species with small and generally well-defined ranges, or restricted species with close relatives that grow in similar but disjunct areas of habitat. In cases where estimates of $PF_{MTSS}$ exceeded estimates of $PF_K$ and $PF_{10}$ by 0.2 or more we visually inspected each model and selected either $P_K$, $PF_{MTSS}$ or neither as the final estimate of proportional fire area (hereafter $PF_{SDM}$).

Given that we were primarily concerned with identifying species that have likely been adversely affected by the 2019-2020 fires, and that knowledge of the distributions of most species within the study area is incomplete, we adopted a conservative approach by selecting the highest value among $PF_{SR}$, $PF_{BR}$ and $PF_{SDM}$ as our species-wide estimate of proportion burnt (hereafter $PF_M$). All species with $PF_M > 0.50$ were then included in the final species list, a total of 816 species. Of these, MaxEnt models were generated for 644 species, which had ≥10 unique specimen locations. To determine the strength of support for inclusion, each species was then classified according to the type and number of specimen and model-based estimates of $PF > 0.50$, such that (1) $= PF_{SR}$ only, (2) $= PF_{BR}$ only, (3) $= PF_{SDM}$ only, (4) $PF_{SR} + PF_{BR}$, (5) $PF_{SR} + PF_{SDM}$, (6) $PF_{BR} + PF_{SDM}$ or (7) $PF_{SR} + PF_{BR} + PF_{SDM}$. The data show that of the 816 species with $PF_M > 0.50$, 649 (80%) were supported by at least two PF estimates, 58 were supported by $PF_{SR}$ alone, 24 by $PF_{BR}$ alone, and 85 by $PF_{SDM}$ alone. For most taxa in the latter three categories, however, alternative PF estimates were usually just below the 0.5 criterion (Supplementary Data 2).

**Identification and traits of fire-affected species.** For all highly burnt species (i.e., $PF_M > 0.50$; $n = 816$) we visually inspected all species distributions and classified each into (1) endemic to the south-eastern mainland study area, (2) near-endemic to study area, with a small proportion (ca. 10% or less) of records occurring elsewhere (usually in central-eastern Queensland or western Victoria), or (3) non-endemic to the study area. We also determined a selection of traits intended to reveal broader physiognomic and floristic impacts of the fires on south-east Australian vegetation and the potential ability of different taxa to recover. First, we assigned all species to one of 16 structural life and growth forms (hereafter life forms) based primarily on morphology: (1) annual or biennial or monocarpic forbs, (2) geophytic forbs (taxa with large underground storage bulbs or tubers but excluding taxa with swollen taproots, tuberoids or fleshy rhizomes), (3) non-geophytic perennial forbs, (4) annual graminoids, (5) perennial graminoids, (6) low subshrubs and shrubs <1 m, (7) medium shrubs 1–3 m, (8) tall shrubs and mallee eucalypts >3–7 m, (9) low trees >7–20 m, (10) medium trees >20–35 m, (11) tall trees >35 m, (12) climbers, (13) epiphytes, (14) low ferns or cycads, (15) tall ferns or cycads, and (16) aquatic species (Supplementary Data 2). Life or growth form analysis based on similar categories have been widely used to classify Australian plant species[81–85] and allowed us to investigate fire impacts on different structural elements of the vegetation. Life form was determined using data provided in the Atlas of Living Australia (ALA; www.ala.org.au), NSW Flora Online (https://plantnet.rbgsyd.nsw.gov.au/), VICFLORA (https://vicflora.rbg.vic.gov.au/), recent taxonomic revisions and specimen sheets where required.

Since a major objective of the paper was to investigate the impact of the fires on relict rainforest species, we also used these data to classify each species according to prevalence in rainforest communities. We used a simple classification: (1) RF++ = primarily or solely occurring in rainforests, (2) RF+ = occurring in rainforests but not restricted to them; also frequently found in adjacent habitats, especially wet sclerophyll forests, 3) RF = sometimes found around the margins of rainforest but more prevalent in other habitats; not typically a rainforest species, and 5) RF− = not a rainforest species. We also noted species predominantly associated with sclerophyll forest or woodland, heathland or shrubland, rocky terrain (including granite tors, cliffs, rock fissures, exposed rocky peaks, outcrops, scree slopes) or steep gullies or gorges in rough terrain (Supplementary Data 2). We also noted a range of other habitat types and on overall habitat description of each species (Supplementary Data 2). Data were sourced primarily from the ALA, NSW Flora Online, VICFLORA and, for rainforest taxa, the Rainforest Plants of Australia Interactive Identification Key and Information System (http://rainforestplantsofaustralia.com/), taxa listed in Department of the Arts, Sport and the Environment and Territories 1992[29] and expert opinion. Descriptions provided on specimen records were also used to determine primary habitat. Data were limited for some species and some taxa may occur in habitats not noted in these sources.

Finally, we used the fire response classification system of Pausas et al.[33] to determine two key fire-related traits for 270 species with $PF_M > 0.5$: (1) resprouting ability, either resprouters (R+) or non-resprouters (R-), and (2) propagule persistence, either propagule persisters (P+) or non-persisters (P-). In this system R+ species are capable of resprouting from vegetative structures such as rhizomes, epicormic buds or lignotubers following 100% canopy scorch, while populations of P+ species persist via a persistent seed bank, serotiny, or pyrogenic flowering. These two traits were then used to classify each plant species into one of four fire response types: (1) R + P + = facultative species that can survive fire by either resprouting or recruiting from a persistent seedbank, (2) R + P− = obligate resprouters that depend on post-fire vegetative regrowth and recruit only after production of new propagules in the inter-fire period, (3) R − P + = obligate seeders, which are post-fire recruitment specialists, and (4) R − P − = non-

persisters, or species that do not survive fire and recover only following dispersal of propagules from unburnt sites. For simplicity, R + P +, R + P − and R − P + species, and where data were partial, R + and P + species, were collectively referred to as fire persisters. Fire response data were sourced primarily from the New South Wales Flora Fire Response Database (supplied by Fire Ecology Unit, NSW Office of Environment and Heritage, Hurstville, NSW 1481) supplemented by additional references and expert opinion (Supplementary Data 3).

**Data analyses.** Statistical analyses were used to investigate relationships among range extent, life form, habitat type and fire impacts across taxa. First, we quantified the skewness of the distribution of maximum range extent ($RE_T$) across taxa endemic to the study area and tested deviation from normality using the Shapiro-Wilk statistic W. We then compared $RE_T$ across rainforest habitat categories (RF ++, RF+, RF and RF−) using simple one-way analysis of variance with data transformed [$y = \log_{10}(x + 1)$] prior to analysis to meet model assumptions of normality and homoscedasticity; post hoc means comparisons were made using Tukey's Honest Significant Difference method. Additional statistical information, including support provided by alternative statistical models is provided in the Supplementary Notes.

We quantified range size ($RE_T$) across different life form categories and tested for group differences using the nonparametric Kruskal–Wallis rank-sum test and post-hoc pairwise Wilcox rank-sum tests with Benjamini and Hochberg's[86] BH adjustment for multiple comparisons. We also aggregated range size data into six broader life form groupings (low shrubs, medium shrubs, tall shrubs, trees, perennial forbs and other, including epiphytes and climbers) and four range classes (extremely and very range restricted with ranges <25 km, range restricted with ranges 25–100 km, moderately widespread with ranges 100–500 km, and very or extremely widespread with ranges >500 km). Row-column contingency was tested using the standard Pearson $\chi^2$ statistic (Supplementary Notes).

The relationship between $RE_T$ and proportion burnt ($PF_M$) across taxa was triangular and hence violated distributional assumptions of linear regression (Supplementary Notes). To quantify this relationship we first tested for an overall difference in $PF_M$ across eight $RE_T$ size classes (<10 km, 10 to <25 km, 25 to <100 km, 100 to <250 km, 250 to <500 km, 500 to <750 km, 750 to <1000 km and 1000 to 1500 km) using the Kruskal–Wallis rank-sum test. We then used quantile regression to characterise the relationship between $RE_T$ and $PF_M$ at 0.10, 0.20....0.90 quantiles ($\tau$) using linear quantile regression models with coefficient confidence intervals estimated using the Koenker rank method. Finally, we compared $PF_M$ across rainforest categories (RF++, RF+, RF and RF−) using the Kruskal–Wallis rank-sum test with post-hoc pairwise group comparisons based on pairwise Wilcox rank-sum tests with the BH adjustment for multiple comparisons. Statistics were performed using the R stats base package version 4.0.1, moments package version 0.14 and the quantreg package version 5.75.

**Reporting summary.** Further information on research design is available in the Nature Research Reporting Summary linked to this article.

## Data availability

Source data and output data that support the results reported in this study are publicly available on the CSIRO Data Access Portal (https://data.csiro.au) at https://doi.org/10.25919/sd7h-ff33. This includes fire hotspot, NVIS and IBRA source datasets downloaded under Creative Commons Attribution 4.0 International Licence and generated fire layers. Data used to generate statistics reported in the paper are available in Supplementary Data 2. Source data are provided with this paper.

## Code availability

The code used to generate data and images reported in this study was written in the open-source programming language R and is publicly available on the CSIRO Data Access Portal (https://data.csiro.au) at https://doi.org/10.25919/sd7h-ff33.

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

## Acknowledgements

The authors thank Liz Tasker, Rod Fensham, Neville Walsh, Phil Gilmour and Pete Richards for assistance with fire response traits. We thank Murray Fagg and curators of the Australian Plant Image Index for providing images.

## Author contributions

L.M.B. and R.G. conceived of the initial research questions. All authors participated in a workshop to identify the scope of the work. R.G., L.M.B. and F.E-V. further developed the scope and structure of the research; D.B., D.C.C, M.C., L.J., A.S-L., B.L., C.G., L.G., T.H., and K.N contributed to development of biogeographic, ecological and taxonomic themes. N.K. led, and R.G. contributed to, coding throughout the study. N.K. and R.G. analysed the hotspot database and N.K., R.G. and L.M.B. developed the fire map. N.K., R.G. and D.A developed the herbarium specimen database. D.A., D.C.C., C.G., M.C., B.L. and K.N. provided taxonomic expertise. R.G., D.A., M.C., L.G. and B.L. developed the plant species trait and life history database. N.K. and R.G. analysed the spatial impact of the fires on vegetation layers and individual taxa. T.H. contributed expertise for Maxent modelling. R.G. developed metrics and the conceptual model reported in the paper. R.G. and F.E.-V. performed statistical analyses. All authors then contributed taxonomic, biogeographic or ecological expertise to interpretation of data. R.G. led, and all authors contributed to, development of first, revised and final versions of the paper.

## Competing interests

The authors declare no competing interests.
