## [Peer Review File · Nature Communications]

Reviewer comments, first round:

Reviewer #1 (Remarks to the Author):

This manuscript combines mapped data (from remote sensing) on the intensity of fires across a very large area of southeastern Australia during 2019–2020 with species occurrence data for plant and fungal taxa from herbaria, to show that a large number of taxa were affected by the fires. This is important in the context that the extent of the fires in a single season is without precedent in at least the last 200 years.

I think that the manuscript would be best if it presented only its strongest data on species distributions, i.e., of native vascular plants. Of the taxa assessed, only these are likely to have been sampled sufficiently across the whole study area (>10 million ha); as acknowledged (line 117), bryophytes and fungi are under-collected compared with vascular plants. I doubt that bryophytes and fungi have been sampled adequately to determine their distributions across all bioregions in the study, or evenly across them. I especially think that the paper would be better to drop fungi from the study. I doubt the low number of fungal taxa identified (196; line 265) within >10 million ha affords much insight; consider that for wood-decay fungi alone, 410 taxa were found within just one ha in a Tasmanian Eucalyptus forest (Gates et al. 2011). Moreover, given the differential effects of fire on different groups of fungi (mycorrhizal, saprotrophic, pathogenic; Semenova-Nelsen et al. 2019, Hansen et al. 2019), for this study to offer insight, all these groups should be represented, and even if they are, I expect that there is low likelihood that the different fungal groups have been sampled evenly across the large study area. I recommend dropping lines 108–125, and replacing them with a sentence or two to state that more comprehensive and even coverage of the distributions of other taxa are needed to evaluate the effects on them, given that some are likely to be sensitive to fires, or key to the recovery of forests from fire (e.g., mycorrhizal fungi). Focusing on native vascular plants remains consistent with the intent of the title (impact of megafires on Australian vegetation).

The authors should give some serious thought to going beyond the point data for individual species (again I recommend only vascular plants) and applying species distribution models to them to overlap these with the remotely-sensed and spatially extensive fire data. Given development of species distribution models is well advanced in Australia (e.g., Fithian et al. 2015, among many), this paper could capitalize on that effort to overlap two spatially extensive datasets. It was unclear to me how geographic ranges (as in Figure 3) had been determined from point data.

The lack of obvious statistical tests (see comments about Table 1), of comparisons among bioregions, among taxa with different range sizes, and among vegetation types, meant that I had little sense of the relative importance of various points made in lines 74 – 107, and lines 126 – 156 (the main “Results” paragraphs). While Figures 1–3 are visually appealing, the lack of any sense of whether differences illustrated in them are significant is a major impediment to having confidence in the results.

I could not tell how (or if) the “Australian Major Vegetation Groups” (per line 60, Table 1) related to authoritative references (especially Keith 2017). Are the classifications quantitative, qualitative? How were they mapped? Whatever is the case, I wanted more information than was given at lines 281–286.

The points made about the particular sensitivity of rain forests to fire, and the damage caused to them during the 2019–2020 fires, at lines 175–177 and 193–198, were made by Kooyman et al. (2020), which should be cited. It may also be worth citing a recent paper (Lindenmayer & Taylor 2020) that also conducted spatial analyses of the 2019–2020 fires in a subset of the area covered in this study.

Other points:

It seemed unnecessary to take a reader through material about algae, bacteria, and protozoans when there were so few taxa identified within >10 million ha (lines 265–266; see typical phylogenetic richness of soil bacteria at a 100 m² scale in Fierer & Jackson 2006).

Table 1: I did not know what the "3" digit in the heading "Broad vegetation type" referred to. I did not know what the superscript "2" referred to in the heading "Bioregional impact". I could not tell what the superscript letters within individual bioregions meant; perhaps they related to statistical significance but, if so, it was not explained in the caption.

Line 90: What is "this region"? The previous sentence refers to four bioregions.

Line 134: "endemism" is the noun, I think.

Line 200: If the work is required "urgently", better defense or an explanation is needed for the urgency. I would not expect evidence of regeneration failure (per line 201) to be determined quickly, for example.

References 37 and 46 are the same one as reference 24.

Figure 2 caption, 3rd to last line. The first of the photos credited should be "E", not "D", I expect.

References

Fierer N, Jackson RB 2006. The diversity and biogeography of soil bacterial communities.

Proceedings of the National Academy of Sciences of the USA 103, 626–631.

Fithian W, Elith J, Hastie T, Keith DA 2015. Bias correction in species distribution models: pooling survey and collection data for multiple species. *Methods in Ecology and Evolution* 6, 424–438.

Gates GM, Mohammed C, Wardlaw T, Ratkowsky DA, Davidson NJ 2011. The ecology and diversity of wood-inhabiting macrofungi in a native *Eucalyptus obliqua* forest of southern Tasmania, Australia. *Fungal Ecology* 4, 56–67.

Hansen PM, Semenova-Nelsen TA, Platt WJ, Sikes BA 2019. Recurrent fires do not affect the abundance of soil fungi in a frequently burned pine savanna. *Fungal Ecology* 42, article 100852.

Keith DA (ed) 2017. *Australian Vegetation*. 3rd edition. Cambridge University Press, Cambridge.

Kooyman RM, Watson J, Wilf P 2020. Protect Australia's Gondwana rainforests. *Science* 367, 1083.

Lindenmayer DB, Taylor C 2020. New spatial analyses of Australian wildfires highlight the need for new fire, resource, and conservation policies. *Proceedings of the National Academy of Sciences of the USA* 117, 12481–12485.

Semenova-Nelsen TA, Platt WJ, Patterson TR, Huffman J, Sikes BA 2019. Frequent fire reorganizes fungal communities and slows decomposition across a heterogeneous pine savanna landscape. *New Phytologist* 224, 916–927.

Reviewer #2 (Remarks to the Author):

The manuscript presents an interesting and "hot" topic which was on the headlines last summer. To my opinion the manuscript needs revision before it may be accepted for publication.

Following are my main comments:

1. The authors need to make clear the meaning of each critical term they are using: large sub-mega fire, megafire, major fire, megadrought
2. The authors use herbarium records for "locating" all taxa affected by the Black Summer Fires. How updated were these records? Did they provide information on population size too? Recent information is important for supporting their statements on fire's impact. Information on population size is critical for deciding whether a taxon is range restricted and has a small population and therefore fire's impact upon the taxon may be more crucial.
3. Hot spot temperatures should be provided in Celsius and not in Kelvin.
4. Correspondence between hotspot temperature and fire intensity should be clearly justified and the relevant scale (intensity classes) should be provided and supported by reference.
5. Fire intervals in the broader area of study should be more stressed if the focus of the research is on fires' impact upon the biota and not just the occurrence of fire. This is critical, because most plant taxa possess adaptation mechanisms against fire, but if they face immaturity risk, then they won't be able to recover.
6. It is better to say that Climate change is predicted to cause extreme events and back it up with references.
7. Although hotspot temperature is used to estimate fire intensity, the latter is not discussed or taken into account in the manuscript.
8. My overall opinion is that it would be preferable to avoid talking about impacts upon organisms,

because this has not been studied. What has been studied is the extent of fires and this has been correlated with herbarium records on existing plant and fungal taxa. Maybe it would be better to focus on species known not to have adaptation mechanisms against fire.

Specific comments

line 103: species burnt - it is not the species but the individuals. Better say species affected by fire

line 112: the taxa mentioned were not recorded but reported for the sites

line 138: large sub mega scale fires???

line 140: potentially sensitive to large sub mega fires ???

lines 160-161: equal or greater importance - why?

lines 195-198 need justification

In Methods they say: historical hot spot data; explanation of the term historical is needed.

Reviewer #3 (Remarks to the Author):

The manuscript presents a spatial analysis of the extent of the "Black summer" megafires in Australia and how they are associated with location occurrence records of plant species in the region. They show that the fires in this season burned large proportions of many vegetation types and plant species' ranges, and imply that this will have serious implications for the recovery and population dynamics of these species.

Although I was interested to read this manuscript, I was seriously disappointed both with the level of analysis presented, and the interpretation and thought that went into it.

In particular, the authors seem to have made no attempt to classify the vegetation types that they study in terms of their existing fire regimes and the degree to which the organisms found there are exposed to fire in the normal course of events. I know that there are detailed and useful datasets available which can provide this information and I reference a few below.

Likewise, the authors only make one passing, throw-away comment (on line 166) of the manuscript about the various different life history strategies organisms have for surviving and persisting in environments with fire. This is key: some life history strategies, such as the reseeded species, will be far less affected than organisms that rely on dispersal in from unburned vegetation patches. I have suggested that the authors read the valuable and interesting work of Pausas and Keeley and others in this regard, and attempt to classify the species that they assess in terms of their sensitivity to intense and large fire events.

Following on from this, I am confused by why they limited their analysis to plants. My understanding of reading the Australian literature is that it is the small mammals and insects whose populations depend on dispersal events to recover from fire that are most affected by extreme and large fire events (see the wonderful set of literature from SW Australia -Luke Kelley and colleagues- on this).

While I feel that studies of this sort are essential for assessing the impacts of these extreme fire events, I do not feel that the data and interpretation presented here add much to the story, and might even give a very wrong impression of the impacts that these fires could have on the ecosystems of SE Australia. I started reading this manuscript expecting that I would find out more about how the Gondwanan relic rainforest systems were affected, and other unique and interesting vegetation elements. I had hoped that this would provide an integrated perspective on the impacts, informed by the rich knowledge of the Australian vegetation and its relationship with fire. But this appears just to be a desktop GIS analysis with hardly any reference to the ecological literature available on the ecosystems in question.

Detailed comments:

Line 33: "firegrounds" is not a term that I am familiar with. Do you mean the size of individual fires or the total area burned? In both cases you would need to compare this to an ecosystem with similar vegetation/climate, otherwise it does not make the point that you are trying to make. In fact, you would first have to demonstrate to me that the extent of area burned is the major unique and problematic factor of the Australian bush fires. I.e. In the introduction I would like to see some ecological explanation about why large extensive fires are dangerous/threatening to biodiversity - for example, my understanding of the Australian fires is that due to the extreme drying beforehand they burned into patches of usually non-flammable vegetation. Likewise because the fires were so extensive there wasn't the usual opportunity for recruitment after the fires.

Line 58: a reference for this could be Archibald et al PNAS 2013 (the largest they recorded was ~5Mha), or Hantson et al IJWF 2016

Line 92: You have got to line 92 of this manuscript without once mentioning that forest fires are a part of the natural history of this part of Australia. All the words that you are using "impacted", "damaged" make it sound as though fires are always necessarily a source of damage and destruction in this ecosystem. I agree that the scale and extent of the fires in 2019/2020 is extreme and I am interested in having you indicate to me how damaging they were. However, I think you need to indicate in the introduction that some ecosystems in this region are adapted to fire and require fire for their healthy functioning, and others are fire-sensitive. Therefore, the impacts of these 2019/2020 fires in these different ecosystems need to be assessed with this in mind.

Line 96: and some (but not all) of this biodiversity is related to the unique fire-dependent ecosystems that exist in this part of Australia.

Line 105: For a species that is fire-dependent it might not necessarily matter if ALL of its occurrence records were burnt in the fire. It would depend on its mode of regeneration after the fire. If they regenerate from resprouting or from fire-stimulated seed release and recruitment then this might not necessarily be a problem - except that the stands might be more even-aged than would be considered healthy. If they recruit via dispersal from unburned areas adjacent to burned patches then of course yes this is devastating for their population dynamics. I am missing any ecological understanding of the situation.

Line 118-120: I would expect here that you should divide the vegetation/habitat that burned up into classes based on their sensitivity to fire and the expected fire return intervals/fire intensity with which they usually burn. I know that this information is available - there are many publications on these topics and several regional and national assessments of Australia's fire regimes. Without information on whether the particular species you are mentioning is adapted to fire or not we cannot assess the relevance of your analysis of how much of its range has burned.

Line 133/134 - ok, but if their ranges are so small then in that case they are likely to have their entire range burn even during non-extreme fire years. i.e. they must have adaptations that allow them to survive fire without having to disperse in again from elsewhere. I suggest you read Juli Pausas' useful work comparing different life -history strategies associated with fire and use this to classify the organisms that you are assessing into "high and low risk". Then your data would be useful.

Line 140 - yes exactly, you are now making the point I just made - well I think you should go ahead and find out before you come and present the data to us.

Line 166 - ok finally: 166 lines into the manuscript you mention that the Australian flora are fire adapted. It is too late. you need to mention this in the introduction. Otherwise you give people

totally the wrong idea. But you should also mention the Australian fauna... are they likewise fire adapted? What adaptations exist in the fauna of this region ?

Line 176-180: When I agreed to review this paper I thought you were going to identify these species, and then assess how affected they were by the fires last year. i.e. Rather than just listing these, make them an integral part of your analysis.

Line 194: Yes, please can you present proper data showing how affected these communities were, as opposed to just lumping them in with the rest of the fire-adapted vegetation.

Line 211: I could be incorrect but I thought that the Modis hotspot data were reported in MW/pixel, not in K?

Line 242: It could be larger or smaller! i.e. one hotspot in one corner of a 2.5km grid cell would be identified as a full 2.5km gridcell burned. I.e. just as much room for errors of commission as errors of omission - in fact, probably more.

Line 247 - surely it is actually the fauna: the small mammals and insects and reptiles that are of most concern here. Many of these organisms have fire recovery strategies linked to dispersal in from the unburned matrix and the unprecedented size of these fires could be seriously problematic for their population processes.

Line 287 - but you didn't group them in terms of their adaptation to fire, or their fire regime characteristics. It would have been so simple to do: you could have used data from Murphey 2014, or Bowman 2013 or any one of a number of other rigorous assessments of fire regimes on the continent.

Bowman, D.M.J.S., Murphy, B.P., Burrows, G.E. and Crisp, M.D., 2012. Fire regimes and the evolution of the Australian biota. *Flammable Australia: Fire regimes, biodiversity and ecosystems in a changing world*, pp.27-47.

Murphy, B.P., Bradstock, R.A., Boer, M.M., Carter, J., Cary, G.J., Cochrane, M.A., Fensham, R.J., Russell-Smith, J., Williamson, G.J. and Bowman, D.M., 2013. Fire regimes of Australia: a pyrogeographic model system. *Journal of Biogeography*, 40(6), pp.1048-1058.

Reviewers' comments:

Reviewer #1 (Remarks to the Author):

Reviewer Comment 1: I think that the manuscript would be best if it presented only its strongest data on species distributions, i.e., of native vascular plants. Of the taxa assessed, only these are likely to have been sampled sufficiently across the whole study area (>10 million ha); as acknowledged (line 117), bryophytes and fungi are under-collected compared with vascular plants. I doubt that bryophytes and fungi have been sampled adequately to determine their distributions across all bioregions in the study, or evenly across them. I especially think that the paper would be better to drop fungi from the study. I doubt the low number of fungal taxa identified (196; line 265) within >10 million ha affords much insight; consider that for wood-decay fungi alone, 410 taxa were found within just one ha in a Tasmanian Eucalyptus forest (Gates et al. 2011). Moreover, given the differential effects of fire on different groups of fungi (mycorrhizal, saprotrophic, pathogenic; Semenova-Nelsen et al. 2019, Hansen et al. 2019), for this study to offer insight, all these groups should be represented, and even if they are, I expect that there is low likelihood that the different fungal groups have been sampled evenly across the large study area. I recommend dropping lines 108–125, and replacing them with a sentence or two to state that more comprehensive and even coverage of the distributions of other taxa are needed to evaluate the effects on them, given that some are likely to be sensitive to fires, or key to the recovery of forests from fire (e.g., mycorrhizal fungi). Focusing on native vascular plants remains consistent with the intent of the title (impact of megafires on Australian vegetation).

Response: The reviewer makes a reasonable argument and correctly notes that cryptogams in general are quite poorly sampled compared to vascular plants. We did point this out in the original draft of the paper and included these taxa mainly to raise the point that certain cryptogams may also be at risk following the fires. However, given the diversity of these taxa it would probably be better to investigate these issues in more detail in a separate piece of work. The paper now focuses solely on vascular plants as noted in the Methods and Results and elsewhere.

Reviewer Comment 2: The authors should give some serious thought to going beyond the point data for individual species (again I recommend only vascular plants) and applying species distribution models to them to overlap these with the remotely-sensed and spatially extensive fire data. Given development of species distribution models is well advanced in Australia (e.g., Fithian et al. 2015, among many), this paper could capitalize on that effort to overlap two spatially extensive datasets. It was unclear to me how geographic ranges (as in Figure 3) had been determined from point data.

Response: This is undoubtedly a fair point and it is true that spatial distribution models are widely, and effectively used in Australia, as elsewhere. We have addressed this issue by generating species distribution models (primarily using MaxEnt) for >7,000 vascular plant species within the study area. Reporting the methods and results and incorporating them into the text and figures comprises a very large part of the revisions made to the manuscript. The full methods are described in the section titled "Estimation of fire impacts and species ranges based on herbarium specimen data and species distribution models" (lines 395-471) and in particular lines 412-471 which outline the approach taken to species distribution modelling. This includes detailed information on the MaxEnt approach, criteria for generating binary spatial distribution maps, and the 13 bioclimatic variables and six soil

and landform variables used as predictor variables – which are all publically available (links are provided to all datasets).

I would like to point out here that there are significant challenges associated with generating and interpreting SDMs for an extremely broad suite of species (here >7,000 taxa). Models can be unreliable for species with few location records (which were numerous in our dataset), and in some cases, such as for certain rare taxa that have realised niches that are fall smaller than their fundamental niches (as predicted by biophysical constraints), predicted ranges can overestimate ranges (in many cases we know this because the actual ranges are quite well understood). Similar problems can arise for species with very disjunct or patchy distributions, especially where the taxonomic identity of some outlying populations is questionable. We also found that models were frequently highly influenced by spatial outliers, which necessitated an additional level of scrutiny of the underlying data and in some cases careful screening of specimen datasheets (some of which had spatial, taxonomic or other errors).

The full methods are now described in the paper, but it is worth mentioning several approaches that we took to both the modelling and incorporating the result into the paper. First, because of the issues noted above, we felt that it was better to include potential species than exclude them based on spurious data. We therefore took a conservative approach to generating final estimates of populations or ranges burnt for each species – in this case retaining any species with >50% burned based on either specimen location records, gridded location records, or SDMs. This is outlined in lines 458-471. Since multiple estimates were involved, we also quantified the support for inclusion of each species – these data show that an additional 85 species (~10%) were included solely on the basis of spatial distribution models, which supports the original reviewer’s concerns. Data for each species, including all estimates, are now available in Supplementary Table S3.

Reviewer Comment 3: The lack of obvious statistical tests (see comments about Table 1), of comparisons among bioregions, among taxa with different range sizes, and among vegetation types, meant that I had little sense of the relative importance of various points made in lines 74 – 107, and lines 126 – 156 (the main “Results” paragraphs). While Figures 1–3 are visually appealing, the lack of any sense of whether differences illustrated in them are significant is a major impediment to having confidence in the results.

Response: Fair comment as it is important for the readers to have confidence in the results of the study. As we have now included extensive investigation of trait (and other data) for >800 species much of the results section has been re-written, and it now includes a range of statistical tests to investigate relationships among vegetation types, proportions of species ranges burnt, life forms, and range extent. These include:

- 1) Distributional aspects of species range extents (normality test etc., lines 160-163),
- 2) Relationship between range size and fire impacts (quantile regression, lines 163-169),
- 3) Differences in range size across life form types affected by the fires (contingency analysis, lines 170-174)
- 4) Relationship between range size across rainforest and non-rainforest species groups (ANOVA, lines 199-201),
- 5) Differences in spatial fire impacts across rainforest and non-rainforest species groups (ANOVA, lines 201-204).

To keep the paper to a reasonable length we did not use statistics to look at bioregional differences across MVGs etc., but these data are simple estimates of total area burnt based on an underlying fire map and since we are not attempting inference across unsampled habitat or areas and these would be of questionable value. The statistics that we did include, however, revealed some interesting insights and certainly improved the paper.

If the reviewers would like to look at the specific tests performed, the underlying data are provided in Supplementary Tables 3 and 5, and the R code can be provided upon request.

Reviewer Comment 4: I could not tell how (or if) the “Australian Major Vegetation Groups” (per line 60, Table 1) related to authoritative references (especially Keith 2017). Are the classifications quantitative, qualitative? How were they mapped? Whatever is the case, I wanted more information than was given at lines 281–286.

Response: Perhaps this detail was lost in the previous version of the paper, but the MVG data (and shape files) are part of the Department of Agriculture, Water and the Environment’s National Vegetation Information System which is a high quality data set appropriate for this work. More information can be found at <http://www.environment.gov.au/land/native-vegetation/national-vegetation-information-system> and this link is provided in the Methods.

Reviewer Comment 5: The points made about the particular sensitivity of rain forests to fire, and the damage caused to them during the 2019–2020 fires, at lines 175–177 and 193–198, were made by Kooyman et al. (2020), which should be cited. It may also be worth citing a recent paper (Lindenmayer & Taylor 2020) that also conducted spatial analyses of the 2019–2020 fires in a subset of the area covered in this study.

Response: Reference added, the sensitivity of rainforests is concerning.

Reviewer Comment 6: It seemed unnecessary to take a reader through material about algae, bacteria, and protozoans when there were so few taxa identified within >10 million ha (lines 265–266; see typical phylogenetic richness of soil bacteria at a 100 m² scale in Fierer & Jackson 2006).

Response: Responded to above; as suggested all of these taxa removed from the paper.

Reviewer Comment 7: Table 1: I did not know what the “3” digit in the heading “Broad vegetation type” referred to. I did not know what the superscript “2” referred to in the heading “Bioregional impact”. I could not tell what the superscript letters within individual bioregions meant; perhaps they related to statistical significance but, if so, it was not explained in the caption.

Response: all these issues have been corrected in the Table

Reviewer Comment 8: Line 90: What is “this region”? The previous sentence refers to four bioregions.

Response: good point, now corrected

Reviewer Comment 9: Line 134: “endemism” is the noun, I think.

Response: as I understand it either endemism or endemicity are Ok, and both are used in the literature. But happy to make this change; endemism is used throughout.

Reviewer Comments 10: line 200: If the work is required “urgently”, better defense or an explanation is needed for the urgency. I would not expect evidence of regeneration failure (per line 201) to be determined quickly, for example.

References 37 and 46 are the same one as reference 24.

Figure 2 caption, 3rd to last line. The first of the photos credited should be “E”, not “D”, I expect.

Response: All corrected

Reviewer #2 (Remarks to the Author):

Reviewer Comment 1. The authors need to make clear the meaning of each critical term that are using: large sub-mega fire, megafire, major fire, megadrought

Response: We agree that these terms were not well defined in the paper and they now have been corrected throughout. We use megafire to refer to any fire >100,000 ha in area.

Reviewer Comment 2. The authors use herbarium records for "locating" all taxa affected by the Black Summer Fires. How updated were these records? Did they provide information on population size too? Recent information is important for supporting their statements on fire's impact. Information on population size is critical for deciding whether a taxon is range restricted and has a small population and therefore fire's impact upon the taxon may be more crucial.

Response: the reviewer raises important points here. We are certainly limited by the nature of herbarium records, and while these do not often state population size, they are the best that are available and are frequently used for distribution modelling etc. However, it is true that age of the records is important, and we corrected for this by excluding herbarium specimens collected prior to 1950. The point concerning range area is also important, given known biases in herbarium data (to which we make reference in the Methods) and this problem was addressed by using species distribution models and other estimation techniques to determine the fraction of each species range that burnt. We also determined the maximum range extent for each species using specimen records, and grouped these by distance, to get a better understanding of how many taxa could be considered range restricted. This information is now provided in Figure 3. Hopefully this addresses the reviewer's concerns.

Reviewer Comment 3. Hot spot temperatures should be provided in Celcius and not in Kelvin.

Response: now provided in the Methods section.

Reviewer Comments 4-5. Correspondence between hotspot temperature and fire intensity should be clearly justified and the relevant scale (intensity classes) should be provided and supported by reference. Although hotspots temperature is used to estimate fire intensity, the latter is not discussed or taken into account in the manuscript.

Response: strictly speaking, we did not deal with intensity, only hotspot temperature. This is now corrected throughout the paper to avoid any confusion.

Reviewer Comment 6. Fire intervals in the broader area of study should be more stressed if the focus of the research is on fires' impact upon the biota and not just the occurrence of fire. This is critical, because most plant taxa possess adaptation mechanisms against fire, but if they face immaturity risk, then they won't be able to recover.

Reviewer Comment 6. Agreed. These considerations, and especially adaptation to fire among the flora, have now been explicitly addressed in the paper (see further responses to the same point by reviewer 3)

Reviewer Comments 7. It is better to say that Climate change is predicted to cause extreme events and back it up with references.

Response: done

Reviewer Comment 8. My overall opinion is that it would be preferable to avoid talking about impacts upon organisms, because this has not been studied. What has been studied is the extent of fires and this has been correlated with herbarium records on existing plant and fungal taxa. Maybe it would be better to focus on species known not to have adaptation mechanisms against fire.

Response: this is a significant comment also made by the third reviewer, who also makes the point that we did not deal with fire impacts in this study. This is true, but two points are worth making. First, information on fire adaptations are not known for a substantial fraction of the species that were in the fire areas. Second, the impacts of the fires on taxa are not necessarily only limited to specific impacts of the fires themselves. For example, the loss of adult plants and replacement by

juvenile seedlings or foliage may have significant consequences for taxa that are considered fire-adapted. Nevertheless, we acknowledge that further investigation of these considerations would add value to the paper. We therefore have added information on fire adaptations (using the fire persister-non persister dichotomy which is based on resprouter-seeder traits) for many of the species in our list (270 taxa), and use this to identify different groups of plants that are likely to respond differently to the fires, which includes taxa which do not survive fire. This information is now provided in Supplementary Table 6 and the associated conceptual model is provided in Figure 3. Discussion of fire traits and the implications for different taxa is provided in the section titled 'Implications for conservation biogeography'.

Reviewer Comments 9: Specific comments

line 103: species burnt - it is not the species but the individuals. Better say species affected by fire

line 112: the taxa mentioned were not recorded but reported for the sites

line 138: large sub mega scale fires???

line 140: potentially sensitive to large sub mega fires ???

lines 160-161: equal or greater importance - why?

lines 195-198 need justification

In Methods they say: historical hot spot data; explanation of the term historical is needed.

Response: all corrected or addressed

Reviewer #3 (Remarks to the Author):

Reviewer Comments 1: Although I was interested to read this manuscript, I was seriously disappointed both with the level of analysis presented, and the interpretation and thought that went into it. In particular, the authors seem to have made no attempt to classify the vegetation types that they study in terms of their existing fire regimes and the degree to which the organisms found there are exposed to fire in the normal course of events. I know that there are detailed and useful datasets available which can provide this information and I reference a few below. Likewise, the authors only make one passing, throw-away comment (on line 166) of the manuscript about the various different life history strategies organisms have for surviving and persisting in environments with fire. This is key: some life history strategies, such as the reseeding species, will be far less affected than organisms that rely on dispersal in from unburned vegetation patches. I have suggested that the authors read the valuable and interesting work of Pausas and Keeley and others in this regard, and attempt to classify the species that they assess in terms of their sensitivity to intense and large fire events.

Response: we agree with the reviewer that these are important considerations. They were not really within the scope of the original paper which was simply intended to describe the pattern and extent of fires, the habitat types within the burnt areas, and to provide a provisional list of taxa potentially affected. However, we have attempted to address these concerns by:

- 1) Providing fire adaptation data for 270 of the species identified in the study using the Pausas reference provided below to classify each on the basis of resprouting ability and propagule persistence (Supplementary Table 6),
- 2) Providing more detailed information on the habitat preferences of each species including rainforest affinity and presence in sclerophyll vegetation, areas of rocky relief and other refugia that might be related to the ability of a species to recover from the fires (Supplementary Table 3),
- 3) Discussing the nature of different taxa affected in terms of range size and endemism (throughout text; Figs. 2-4; Supplementary Tables 3-5).

- 4) Using a simple conceptual model to discriminate between species likely and unlikely to persist following fire (Figure 3 and text), and
- 5) Identify taxa that are most at risk of demographic decline and map the locations where burnt and unburnt populations occur (Figure 4 and Supplementary Figure 3).

These revisions have required essentially a full re-write of the sections of the paper that dealt with impacts of the fires on specific taxa, which are now provided in lines 206-257 of the text.

Reviewer Comments 2: Following on from this, I am confused by why they limited their analysis to plants. My understanding of reading the Australian literature is that it is the small mammals and insects whose populations depend on dispersal events to recover from fire that are most affected by extreme and large fire events (see the wonderful set of literature from SW Australia -Luke Kelley and colleagues- on this).

Response: Investigation of impacts on animals is outside the scope of the paper. Based on the comments of other reviewers we have focused on the vascular plants, which still involves investigation and modelling of over 7,000 taxa.

Reviewer Comments 3: While I feel that studies of this sort are essential for assessing the impacts of these extreme fire events, I do not feel that the data and interpretation presented here add much to the story, and might even give a very wrong impression of the impacts that these fires could have on the ecosystems of SE Australia. I started reading this manuscript expecting that I would find out more about how the Gondwanan relic rainforest systems were affected, and other unique and interesting vegetation elements. I had hoped that this would provide an integrated perspective on the impacts, informed by the rich knowledge of the Australian vegetation and its relationship fire. But this appears just to be a desktop GIS analysis with hardly any reference to the ecological literature available on the ecosystems in question.

Response: Again, while we did discuss some of these themes in the original manuscript, we were really intending for the study to briefly outline the development and scale of the fires, the vegetation communities impacted, and some taxa affected in the immediate aftermath of the fires, all within a relatively short paper typical of *Nature Communications*. We hope that the additional, more detailed investigation of the life forms, habitat preferences, range sizes and fire responses provided in the paper, notwithstanding the limitations imposed by the length of the paper, address these concerns.

Reviewer Comments 4. Line 33: "firegrounds" is not a term that I am familiar with. Do you mean the size of individual fires or the total area burned? In both cases you would need to compare this to an ecosystem with similar vegetation/climate, otherwise it does not make the point that you are trying to make. In fact, you would first have to demonstrate to me that the extent of area burned is the major unique and problematic factor of the Australian bush fires. I.e. In the introduction I would like to see some ecological explanation about why large extensive fires are dangerous/threatening to biodiversity - for example, my understanding of the Australian fires is that due to the extreme drying beforehand they burned into patches of usually non-flammable vegetation. Likewise because the fires were so extensive there wasn't the usual opportunity for recruitment after the fires.

Response: First, we don't use the term fireground now, as it can be used in a different context in other fields. We are really dealing with fire areas, and this is now explicitly stated.

In the comment above, and here, the reviewer notes or implies that the original paper might give the wrong impression of the impact of the fires on the Australian flora. I think that they are touching on an important issue here and one that I have subsequently given serious thought to. Having now looked at the specific habitat types affected (mainly sclerophyll vegetation which burn reasonably

frequently), the fire responses of the species that fall within the burnt areas (mainly resprouters and reseeder), and the range sizes of affected taxa (many very small and range-restricted), in the authors' view the data indicate that, for many species, the 2019-2020 fires were not necessarily unprecedented. In fact, the evidence suggests that most species, including rare and range-restricted ones, should recover. We made this point briefly in the original paper, but now we discuss it in much more detail, including in relation to a conceptual model (Fig. 3) that discriminates between different taxa based on the factors mentioned above. The specific conclusion that most taxa are likely to recover from the fires is now mentioned in the abstract, which hopefully addresses the reviewer's concern.

The reviewer also raises the issue of whether the mere size of the burnt areas is also a unique or interesting feature of the fires. We now deal with this in more detail by explicitly discussing the implications of the fires for widespread taxa, and particularly the demographic consequences of widespread loss of adult plants in species where this would not occur during normal fire seasons, using the example of myrtle rust. This is now addressed in lines 233-243.

Reviewer Comments 4. Line 58: a reference for this could be Archibald et al PNAS 2013 (the largest they recorded was ~5Mha), or Hantson et al IJWF 2016

Response: this section has been rewritten

Overall response to remainder of reviewer comments

The reviewer has made many comments below indicating that in this paper we needed to go into far more detail concerning the response of different species to fire, the importance of fire in some Australian ecosystems, the actual impact and level of risk to different taxa, and discriminating between fire-sensitive and non-fire sensitive ecosystems. Many of these arguments are reasonable but were obviously outside the scope of the original paper, although as the reviewer notes we did in fact address many of the points that they raised towards the end of the paper. Our intention was not to provide detailed species-level information for all taxa, especially fire response data, which is by no means as easy to obtain as the reviewer suggests. Indeed, studies are now being conducted across NSW to determine the response of many taxa to the fires. I think that the main bone of contention for the reviewer is that we did not make it clear enough that fires are a natural part of these ecosystems and that most species are likely to recover. We don't want to give this impression.

To address these concerns we have rewritten the paper and included fire response data for a very large subset of the taxa (270 species). This involved use of the most up to date fire response database that we are aware of, combined with additional published sources and, for a number of taxa lacking published data, expert opinion. We also determined the basic habitat types in which species occur, and in particular sclerophyll vegetation which is fire prone. Thus, our conclusions are now data-driven. These data show, as the reviewer suggests, that most of the species within the study area are likely to recover from the fires, with a few exceptions, and that fires occur regularly in these ecosystems. These text of the paper now contains:

- 1) The resilience of much of the vegetation to the fires is mentioned in the abstract;
- 2) The history of fires in the study area is provided in the 'extent of fires in vegetation communities' section;
- 3) That rainforest taxa comprise only a small part of the affected flora is stated in lines 137-140;
- 4) The fact that widespread species had smaller fractions of their ranges burnt is mentioned in lines 164-170 and 200-203;

- 5) The ability of the majority of taxa, especially species occurring in sclerophyll vegetation, to recover via resprouting or regenerating from seed, is detailed in lines 217-232;
- 6) Specific examples of taxa that are likely to be fire-sensitive, and why, are provided in lines 244-257;
- 7) Specific reasons why certain widespread taxa might be subjected to novel post-fire stresses are now mentioned in lines 233-243;
- 8) The fact that we are optimistic about the prospects for most species is the first sentence in the Conclusions section.

We hope that these revisions and additions to the paper satisfy the concerns of the reviewer. Below, we respond briefly to any additional comments.

Reviewer Comment 5. Line 92: You have got to line 92 of this manuscript without once mentioning that forest fires are a part of the natural history of this part of Australia. All the words that you are using "impacted", "damaged" make it sound as though fires are always necessarily a source of damage and destruction in this ecosystem. I agree that the scale and extent of the fires in 2019/2020 is extreme and I am interested in having you indicate to me how damaging they were. However, I think you need to indicate in the introduction that some ecosystems in this region are adapted to fire and require fire for their healthy functioning, and others are fire-sensitive. Therefore, the impacts of these 2019/2020 fires in these different ecosystems need to be assessed with this in mind. Line 96: and some (but not all) of this biodiversity is related to the unique fire-dependent ecosystems that exist in this part of Australia.

Response: the scope of the paper has been expanded to address this comment; fire responses and vegetation types are now discussed at length.

Reviewer Comment 6. Line 105: For a species that is fire-dependent it might not necessarily matter if ALL of its occurrence records were burnt in the fire. It would depend on its mode of regeneration after the fire. If they regenerate from resprouting or from fire-stimulated seed release and recruitment then this might not necessarily be a problem - except that the stands might be more even-aged than would be considered healthy. If they recruit via dispersal from unburned areas adjacent to burned patches then of course yes this is devastating for their population dynamics. I am missing any ecological understanding of the situation.

Response: the scope of the paper has been expanded to specifically discriminate between different fire response types across taxa; data are provided in Supplementary Table 6 and specific examples of each group of taxa, or relevant references, are provided in the *Implications for conservation biogeography* section.

Reviewer Comment 7. Line 118-120: I would expect here that you should divide the vegetation/habitat that burned up into classes based on their sensitivity to fire and the expected fire return intervals/fire intensity with which they usually burn. I know that this information is available - there are many publications on these topics and several regional and national assessments of Australia's fire regimes. Without information on whether the particular species you are mentioning is adapted to fire or not we cannot assess the relevance of your analysis of how much of its range has burned.

Response: Given the very large number of vegetation types and habitats affected by the fires it would be far beyond the scope of this paper, and indeed likely any single paper, to fully address this issue, even if data were available for all species and communities. Nevertheless we have addressed it to the extent possible, and the central point concerning fire adaptedness of the vegetation is now thoroughly made and backed up with data.

Reviewer Comment 8: Line 133/134 - ok, but if their ranges are so small then in that case they are likely to have their entire range burn even during non-extreme fire years. i.e. they must have adaptations that allow them to survive fire without having to disperse in again from elsewhere. I suggest you read Juli Pausas' useful work comparing different life-history strategies associated with fire and use this to classify the organisms that you are assessing into "high and low risk". Then your data would be useful.

Response: the point was made in the paper concerning the implications for rare or range-restricted species, but perhaps in not enough detail. We have now addressed this by using Pausas' framework to classify species according to fire sensitivity and integrated this into a deeper discussion of the interrelated roles of range size, fire sensitivity, and life history in the Results.

Reviewer Comment 9. Line 140 - yes exactly, you are now making the point I just made - well I think you should go ahead and find out before you come and present the data to us.

Response: OK, this was done and it now forms a key part of the paper. However, for the reasons stated above I note that this was not exactly the simple task it was suggested to be. Hopefully the detailed investigation of a subset of species and better developed discussion of fire responses that we provide will suffice.

Reviewer Comment 10. Line 166 - ok finally: 166 lines into the manuscript you mention that the Australian flora are fire adapted. It is too late. you need to mention this in the introduction. Otherwise you give people totally the wrong idea. But you should also mention the Australian fauna... are they likewise fire adapted? What adaptations exist in the fauna of this region ?

Response: The first point is now made in the abstract to avoid confusion. The role of fire with respect to fauna is outside the scope of this paper.

Reviewer Comment 11: Line 176-180: When I agreed to review this paper I thought you were going to identify these species, and then assess how affected they were by the fires last year. i.e. Rather than just listing these, make them an integral part of your analysis.

Response: We have certainly taken this major concern on board and have worked very hard to include the information that the reviewer has requested. I would urge the reviewer to look at the data presented in Supplementary Table 6 and the associated analyses and see if their concerns have been addressed.

Reviewer Comment 12: Line 194: Yes, please can you present proper data showing how affected these communities were, as opposed to just lumping them in with the rest of the fire-adapted vegetation.

Response: now included as requested.

Reviewer Comment 13: Line 211: I could be incorrect but I thought that the Modis hotspot data were reported in MW/pixel, not in K?

Response: we just use hotspot temperature data here.

Reviewer Comment 14: Line 242: It could be larger or smaller! i.e. one hotspot in one corner of a 2.5km grid cell would be identified as a full 2.5km gridcell burned. I.e. just as much room for errors of commission as errors of omission - in fact, probably more.

Response: True, and the implications of the modelling approach is mentioned in the Methods.

Reviewer Comment 14: Line 247 - surely it is actually the fauna: the small mammals and insects and reptiles that are of most concern here. Many of these organisms have fire recovery strategies linked to dispersal in from the unburned matrix and the unprecedented size of these fires could be seriously problematic for their population processes.

Response: again, we can't disagree – the implications for animals are probably a lot more significant than they are for most plants. But this is not within the scope of this paper.

Reviewer Comment 14: Line 287 - but you didn't group them in terms of their adaptation to fire, or their fire regime characteristics. It would have been so simple to do: you could have used data from Murphey 2014, or Bowman 2013 or any one of a number of other rigorous assessments of fire regimes on the continent.

Bowman, D.M.J.S., Murphy, B.P., Burrows, G.E. and Crisp, M.D., 2012. Fire regimes and the evolution of the Australian biota. *Flammable Australia: Fire regimes, biodiversity and ecosystems in a changing world*, pp.27-47.

Murphy, B.P., Bradstock, R.A., Boer, M.M., Carter, J., Cary, G.J., Cochrane, M.A., Fensham, R.J., Russell-Smith, J., Williamson, G.J. and Bowman, D.M., 2013. Fire regimes of Australia: a pyrogeographic model system. *Journal of Biogeography*, 40(6), pp.1048-1058.

Response: as noted above, this comment has been addressed as comprehensively as is possible, and now comprises a large part of the revised manuscript.

Reviewer comments, second round:

Reviewer #1 (Remarks to the Author):

This is a substantially revised manuscript that combines mapped data (from remote sensing) on the intensity of fires across a very large area of southeastern Australia during 2019–2020 with vascular plant species distribution data (modelled from herbarium specimens), to show that a large number of taxa were affected by the fires. It also evaluates those species in terms of their life history responses to fires, and concludes most plant populations will recover from fires of a scale and severity not experienced in at least 200 years, but the few rain forest taxa affected may not. I appreciated the authors' thorough response to my and other reviewers' and the editor's comments. I like the new "Impacts on vascular plant taxa" section (line 103) and the new "Implications for conservation biogeography" section (line 155).

My only substantial comment is about plant species' range extents – lines 406, 408–410. Why is this one-dimensional (presumably latitudinal) only? Why isn't it areal extent (i.e., km²; cf. Figures 3a, 3b and 3c, Figures 4a, 4b)? I don't think a one-dimensional delineation of range extent is satisfactory.

I'd like to see posthoc tests of differences in Figures 3b and 3c.

Minor additional comments:

Title: Can fires really have an "impact ... on biogeography"? I don't think so (see also line 37). I'm not even sure that "impact on" is the correct term in the context of conservation. I wonder if "Implications of the 2019–2020 megafires for the conservation of Australian vegetation" might be a better title (cf. line 155). "Biogeography" isn't mentioned in the abstract and perhaps it should be (cf. line 155).

Line 34: Is the term "fireground" (defined in Websters as "an area in which fire-fighting operations are carried on") apt here? Would this be better as "The area burned was almost an order...?"

Line 37: I found this sentence strange. It's also the inverse of the title. How can biogeography and taxonomy have an "impact on the fires"? Might this work better as "In this paper, we quantify the impact of the 2019–2020 megafires across southeastern continental Australia on vascular plant taxa, and their associated vegetation types and biogeography, using a continental fire layer and geocoded species occurrence data"?

Line 43: I think this sentence would be better if it began with the clause "We developed a continent-wide..." (of line 44), but if you must keep it as it is, then insert a comma after "2020".

Lines 71 and 72: The sudden shift from Mha (the rest of the paragraph, including <1 Mha at lines 64 and 65) to 103 ha is strange – I recommend keeping to Mha for the whole paragraph.

Line 381: should be 700,000, I expect.

Line 475: "Queensland" in full – most international readers won't be familiar with the acronym.

Figure 3c: Include the interpretation of all acronyms on the x-axis in the caption – it's too much to expect a reader to be able to decode it.

Reviewer #2 (Remarks to the Author):

My opinion is that you have met most of the reviewers' comments and the manuscript is much improved. However, I would advise you to carefully read the manuscript again, correct several phrasing errors, e.g. line 175 threated ranges (you mean threatened), line 222 ligotubers (you mean lignotubers) etc.

In addition to these minor issues, I suggest that you avoid terms that are not correct, e.g. highly endemic taxa... The taxa are either endemic or not...What is the meaning of highly endemic?

line 207 plant species regenerate and communities recover

line 141, shrubs and trees are not life forms but growth forms. Change the expression everywhere and also in the graphs

You should also provide details which nomenclature you are following

Response to Reviewer Comments

Reviewer #1 (Remarks to the Author):

This is a substantially revised manuscript that combines mapped data (from remote sensing) on the intensity of fires across a very large area of southeastern Australia during 2019–2020 with vascular plant species distribution data (modelled from herbarium specimens), to show that a large number of taxa were affected by the fires. It also evaluates those species in terms of their life history responses to fires, and concludes most plant populations will recover from fires of a scale and severity not experienced in at least 200 years, but the few rain forest taxa affected may not. I appreciated the authors' thorough response to my and other reviewers' and the editor's comments. I like the new "Impacts on vascular plant taxa" section (line 103) and the new "Implications for conservation biogeography" section (line 155).

Response: I am glad that the reviewer is happy with the revisions made to the paper.

Reviewer Comment: My only substantial comment is about plant species' range extents – lines 406, 408–410. Why is this one-dimensional (presumably latitudinal) only? Why isn't it areal extent (i.e., km²; cf. Figures 3a, 3b and 3c, Figures 4a, 4b)? I don't think a one-dimensional delineation of range extent is satisfactory.

Response: the reviewer makes a point that we have considered in some detail before settling on range extent as the measure that we used to quantify species biogeographical extent. The problem with area is that it is very difficult to quantify for species with patchy distributions, for example a species may have three widely scattered but small populations, and thus have a small total range area but large range extent. From a biogeographical standpoint it was of more interest to us whether species with very large range extents (in the hundreds of kms) were among our list of species >50% burnt, because such species would only be affected in this way when multiple megafires occurred in the same season, such as 2019-2020. We think that this provides useful information on the unprecedented impact of the fires on SE Australian vegetation. Given that this comment would require revision of most of the manuscript, we would like to retain our original focus on range extent. However, future work could refine the spatial models and delve into range area data more comprehensively.

Comment: I'd like to see posthoc tests of differences in Figures 3b and 3c.

Response: This is reasonable; statistical tests of group differences and have now been added to Figures 2g and 3c and plots in the Supplementary Notes. We performed a global test of significance on data in figure 3b but not post hoc tests as we did not use statistical inference to test for differences among range categories – appropriate analyses are included in the Supplementary Notes.

Minor additional comments:

Comment: Title: Can fires really have an "impact ... on biogeography"? I don't think so (see also line 37). I'm not even sure that "impact on" is the correct term in the context of conservation. I wonder if "Implications of the 2019–2020 megafires for the conservation of Australian vegetation" might be a

better title (cf. line 155). “Biogeography” isn’t mentioned in the abstract and perhaps it should be (cf. line 155).

Response: I can see what the reviewer is saying here, although I think the point is arguable. However, to avoid confusing the reader, we have changed the title to read:

“Implications of the 2019-2020 Megafires for the Biogeography and Conservation of Australian Vegetation”

- If the Editor would like to retain ‘Impacts’ in the title please let me know

Comment: Line 34: Is the term “fireground” (defined in Webster’s as “an area in which fire-fighting operations are carried on”) apt here? Would this be better as “The area burned was almost an order...”?

Response: yes, thank you, I thought that we had removed reference to ‘firegrounds’ throughout the paper – and this has now been done.

Comment: Line 37: I found this sentence strange. It’s also the inverse of the title. How can biogeography and taxonomy have an “impact on the fires”? Might this work better as “In this paper, we quantify the impact of the 2019–2020 megafires across southeastern continental Australia on vascular plant taxa, and their associated vegetation types and biogeography, using a continental fire layer and geocoded species occurrence data”?

Response: the abstract has been mostly re-written to reduce its length but we have included the ideas contained in the suggested paragraph as much as possible.

Comment: Line 43: I think this sentence would be better if it began with the clause “We developed a continent-wide...” (of line 44), but if you must keep it as it is, then insert a comma after “2020”.

Response: The sentence has been restructured largely as suggested, which has certainly helped its flow.

Comment: Lines 71 and 72: The sudden shift from Mha (the rest of the paragraph, including <1 Mha at lines 64 and 65) to 103 ha is strange – I recommend keeping to Mha for the whole paragraph.

Response: yes, fair enough. The suggested changes have been made.

Comment: Line 381: should be 700,000, I expect.

Response: Yes this is true, correction made.

Comment: Line 475: “Queensland” in full – most international readers won’t be familiar with the acronym.

Response: Correction made

Comment: Figure 3c: Include the interpretation of all acronyms on the x-axis in the caption – it’s too much to expect a reader to be able to decode it.

Response: Correction made

Reviewer #2 (Remarks to the Author):

Comment: My opinion is that you have met most of the reviewers' comments and the manuscript is much improved. However, I would advise you to carefully read the manuscript again, correct several phrasing errors, e.g. line 175 threated ranges (you mean threatened), line 222 ligotubers (you mean lignotubers) etc.

Response: the paper has been thoroughly re-read to hopefully remove any spelling or other errors.

Comment: In addition to these minor issues, I suggest that you avoid terms that are not correct, e.g. highly endemic taxa... The taxa are either endemic or not...What is the meaning of highly endemic?

Response: the reviewer is correct and the term has been removed.

Comment: line 207 plant species regenerate and communities recover

Response: the sentence has been altered to include the term 'regenerate'

Response: line 141, shrubs and trees are not life forms but growth forms. Change the expression everywhere and also in the graphs

Response: I think that these terms are used interchangeably in the ecological literature, and the concept of life forms similar to the ones reported in the paper have been used elsewhere. However, to clarify this issue we now state in the methods that the categories include life form and growth form traits, which we henceforth refer to as life forms.

Comment: You should also provide details which nomenclature you are following

Response: this is in the Methods section.